# Stochastic Approximation Approaches to Group Distributionally Robust Optimization

**Lijun Zhang**[1,2],    **Peng Zhao**[1],    **Zhen-Hua Zhuang**[1],    **Tianbao Yang**[3],    **Zhi-Hua Zhou**[1]

[1]National Key Laboratory for Novel Software Technology, Nanjing University, Nanjing, China
[2]Peng Cheng Laboratory, Shenzhen 518055, China
[3]Department of Computer Science and Engineering, Texas A&M University, College Station, USA
{zhanglj, zhaop, zhuangzh, zhouzh}@lamda.nju.edu.cn, tianbao-yang@tamu.edu

## Abstract

This paper investigates group distributionally robust optimization (GDRO), with the purpose to learn a model that performs well over $m$ different distributions. First, we formulate GDRO as a stochastic convex-concave saddle-point problem, and demonstrate that stochastic mirror descent (SMD), using $m$ samples in each iteration, achieves an $O(m(\log m)/\epsilon^2)$ sample complexity for finding an $\epsilon$-optimal solution, which matches the $\Omega(m/\epsilon^2)$ lower bound up to a logarithmic factor. Then, we make use of techniques from online learning to reduce the number of samples required in each round from $m$ to 1, keeping the same sample complexity. Specifically, we cast GDRO as a two-players game where one player simply performs SMD and the other executes an online algorithm for non-oblivious multi-armed bandits. Next, we consider a more practical scenario where the number of samples that can be drawn from each distribution is different, and propose a novel formulation of weighted GDRO, which allows us to derive *distribution-dependent* convergence rates. Denote by $n_i$ the sample budget for the $i$-th distribution, and assume $n_1 \geq n_2 \geq \cdots \geq n_m$. In the first approach, we incorporate non-uniform sampling into SMD such that the sample budget is satisfied in expectation, and prove that the excess risk of the $i$-th distribution decreases at an $O(\sqrt{n_1 \log m}/n_i)$ rate. In the second approach, we use mini-batches to meet the budget exactly and also reduce the variance in stochastic gradients, and then leverage stochastic mirror-prox algorithm, which can exploit small variances, to optimize a carefully designed weighted GDRO problem. Under appropriate conditions, it attains an $O((\log m)/\sqrt{n_i})$ convergence rate, which almost matches the optimal $O(\sqrt{1/n_i})$ rate of only learning from the $i$-th distribution with $n_i$ samples.

## 1  Introduction

In the classical statistical machine learning, our goal is to minimize the risk with respect to a *fixed* distribution $\mathcal{P}_0$ [Vapnik, 2000], i.e.,

$$\min_{\mathbf{w} \in \mathcal{W}} \left\{ R_0(\mathbf{w}) = \mathrm{E}_{\mathbf{z} \sim \mathcal{P}_0}\left[\ell(\mathbf{w}; \mathbf{z})\right] \right\}, \tag{1}$$

where $\mathbf{z} \in \mathcal{Z}$ is a sample drawn from $\mathcal{P}_0$, $\mathcal{W}$ denotes a hypothesis class, and $\ell(\mathbf{w}; \mathbf{z})$ is a loss measuring the prediction error of model $\mathbf{w}$ on $\mathbf{z}$. During the past decades, various algorithms have been developed to optimize (1), and can be grouped in two categories: sample average approximation (SAA) and stochastic approximation (SA) [Kushner and Yin, 2003]. In SAA, we minimize an empirical risk defined as the average loss over a set of samples drawn from $\mathcal{P}_0$, and in SA, we directly solve the original problem by using stochastic observations of the objective $R_0(\cdot)$.

37th Conference on Neural Information Processing Systems (NeurIPS 2023).

However, a model that trained over a single distribution may lack robustness in the sense that (i) it could suffer high error on minority subpopulations, though the average loss is small; (ii) its performance could degenerate dramatically when tested on a different distribution. Distributionally robust optimization (DRO) provides a principled way to address those limitations by minimizing the worst-case risk in a neighborhood of $\mathcal{P}_0$ [Ben-Tal et al., 2013]. Recently, it has attracted great interest in optimization [Shapiro, 2017], statistics [Duchi and Namkoong, 2021], operations research [Duchi et al., 2021], and machine learning [Hu et al., 2018, Jin et al., 2021, Agarwal and Zhang, 2022]. In this paper, we consider an emerging class of DRO problems, named as Group DRO (GDRO) which optimizes the maximal risk over a finite number of distributions [Oren et al., 2019, Sagawa et al., 2020]. Mathematically, GDRO can be formulated as a minimax stochastic problem:

$$\min_{\mathbf{w} \in \mathcal{W}} \max_{i \in [m]} \left\{ R_i(\mathbf{w}) = \mathrm{E}_{\mathbf{z} \sim \mathcal{P}_i} \big[ \ell(\mathbf{w}; \mathbf{z}) \big] \right\} \tag{2}$$

where $\mathcal{P}_1, \ldots, \mathcal{P}_m$ denote $m$ distributions. A motivating example is federated learning, where a centralized model is deployed at multiple clients, each of which faces a (possibly) different data distribution [Mohri et al., 2019].

Supposing that samples can be drawn from all distributions freely, we develop efficient SA approaches for (2), in favor of their light computations over SAA methods. As elaborated by Nemirovski et al. [2009, §3.2], we can cast (2) as a stochastic convex-concave saddle-point problem:

$$\min_{\mathbf{w} \in \mathcal{W}} \max_{\mathbf{q} \in \Delta_m} \left\{ \phi(\mathbf{w}, \mathbf{q}) = \sum_{i=1}^{m} q_i R_i(\mathbf{w}) \right\} \tag{3}$$

where $\Delta_m = \{ \mathbf{q} \in \mathbb{R}^m : \mathbf{q} \geq 0, \sum_{i=1}^{m} q_i = 1 \}$ is the $(m-1)$-dimensional simplex, and then solve (3) by their mirror descent stochastic approximation method, namely stochastic mirror descent (SMD). In fact, several recent studies have adopted this (or similar) strategy to optimize (3). But, unfortunately, we found that existing results are unsatisfactory because they either deliver a loose sample complexity [Sagawa et al., 2020], suffer subtle dependency issues in their analysis [Haghtalab et al., 2022, Soma et al., 2022], or hold only in expectation [Carmon and Hausler, 2022].

As a starting point, we first provide a routine application of SMD to (3), and discuss the theoretical guarantees. In each iteration, we draw 1 sample from every distribution to construct unbiased estimators of $R_i(\cdot)$ and its gradient, and then update both $\mathbf{w}$ and $\mathbf{q}$ by SMD. The proposed method achieves an $O(\sqrt{(\log m)/T})$ convergence rate in expectation and with high probability, where $T$ is the number of iterations. As a result, we obtain an $O(m(\log m)/\epsilon^2)$ sample complexity for finding an $\epsilon$-optimal solution of (3), which matches the $\Omega(m/\epsilon^2)$ lower bound [Soma et al., 2022, Theorem 5] up to a logarithmic factor, and tighter than the $O(m^2(\log m)/\epsilon^2)$ bound of Sagawa et al. [2020] by an $m$ factor. While being straightforward, this result seems *new* for GDRO.

Then, we proceed to reduce the number of samples used in each iteration from $m$ to 1. We remark that a naive uniform sampling over $m$ distributions does not work well, and yields a worse sample complexity [Sagawa et al., 2020]. As an alternative, we borrow techniques from online learning with stochastic observations, and explicitly tackle the *non-oblivious* nature of the online process, which distinguishes our method from that of Soma et al. [2022]. Specifically, we use SMD to update $\mathbf{w}$, and Exp3-IX, an algorithm for non-oblivious multi-armed bandits (MAB) [Neu, 2015], with stochastic rewards to update $\mathbf{q}$. In this way, our algorithm only needs 1 sample in each round and attains an $O(\sqrt{m(\log m)/T})$ convergence rate, implying the same $O(m(\log m)/\epsilon^2)$ sample complexity.

Next, we investigate a more practical and challenging scenario in which there are different budgets of samples that can be drawn from each distribution, a natural phenomenon encountered in learning with imbalanced data [Amodei et al., 2016]. Let $n_i$ be the sample budget of the $i$-th distribution, and without loss of generality, we assume that $n_1 \geq n_2 \geq \cdots \geq n_m$. Now, the goal is not to attain the optimal sample complexity, but to reduce the risk on all distributions as much as possible, under the budget constraint. For GDRO with different budgets, we develop two SA approaches based on non-uniform sampling and mini-batches, respectively.

In each iteration of the first approach, we draw 1 sample from every $\mathcal{P}_i$ with probability $n_i/n_1$, and then construct stochastic gradients to perform mirror descent. In this way, the budget will be satisfied in expectation after $n_1$ rounds. To analyze its performance, we propose a novel formulation of weighted GDRO, which weights each risk $R_i(\cdot)$ in (3) by a scale factor $p_i$. Then, our algorithm can be regarded as SMD for an instance of weighted GDRO. With the help of scale factors, we

demonstrate that the proposed algorithm enjoys *distribution-dependent* convergence in the sense that it converges faster for distributions with more samples. In particular, the excess risk on distribution $\mathcal{P}_i$ reduces at an $O(\sqrt{n_1 \log m}/n_i)$ rate, and for $\mathcal{P}_1$, it becomes $O(\sqrt{(\log m)/n_1})$, which almost matches the optimal $O(\sqrt{1/n_1})$ rate of learning from a single distribution with $n_1$ samples.

On the other hand, for distribution $\mathcal{P}_i$ with budget $n_i < n_1$, the above $O(\sqrt{n_1 \log m}/n_i)$ rate is worse than the $O(\sqrt{1/n_i})$ rate obtained by learning from $\mathcal{P}_i$ alone. In shape contrast with this limitation, our second approach yields nearly optimal convergence rates for *multiple* distributions across a large range of budgets. To meet the budget constraint, it runs for $\bar{n} \leq n_m$ rounds, and in each iteration, draws a mini-batch of $n_i/\bar{n}$ samples from every distribution $\mathcal{P}_i$. As a result, (i) the budget constraint is satisfied *exactly*; (ii) for distributions with a larger budget, the associated risk function can be estimated more accurately, making the variance of the stochastic gradient smaller. To benefit from the small variance, we leverage stochastic mirror-prox algorithm [Juditsky et al., 2011], instead of SMD, to update our solutions, and again make use of the weighted GDRO formulation to obtain distribution-wise convergence rates. Theoretical analysis shows that the excess risk converges at an $O((\frac{1}{n_m} + \frac{1}{\sqrt{n_i}}) \log m)$ rate for each $\mathcal{P}_i$. Thus, we obtain a nearly optimal $O((\log m)/\sqrt{n_i})$ rate for distributions $\mathcal{P}_i$ with $n_i \leq n_m^2$, and an $O((\log m)/n_m)$ rate otherwise. Note that the latter rate is as expected since the algorithm only updates $O(n_m)$ times.

**Related work**  We briefly discuss the related works on the GDRO problem in (2)/(3) and will review the traditional DRO problem in Appendix A. Sagawa et al. [2020] have applied SMD [Nemirovski et al., 2009] to (3), but only obtain a sub-optimal sample complexity. In the sequel, Haghtalab et al. [2022] and Soma et al. [2022] have tried to improve the sample complexity by reusing samples and applying techniques from MAB respectively, but their analysis suffers dependency issues. Carmon and Hausler [2022, Proposition 2] successfully established an $O(m(\log m)/\epsilon^2)$ sample complexity by combining SMD and gradient clipping, but their result holds only in expectation. To deal with heterogeneous noise in different distributions, Agarwal and Zhang [2022] propose a variant of GDRO named as minimax regret optimization (MRO), which replaces the risk with "excess risk".

## 2  SA Approaches to GDRO

In this section, we provide two efficient SA approaches for GDRO, which are equipped with the same sample complexity, but with different number of samples used in each round ($m$ versus 1).

### 2.1  Preliminaries

First, we state the general setup of mirror descent [Nemirovski et al., 2009]. We equip the domain $\mathcal{W}$ with a distance-generating function $\nu_w(\cdot)$, which is 1-strongly convex with respect to certain norm $\|\cdot\|_w$. We define the Bregman distance associated with $\nu_w(\cdot)$ as $B_w(\mathbf{u}, \mathbf{v}) = \nu_w(\mathbf{u}) - [\nu_w(\mathbf{v}) + \langle \nabla \nu_w(\mathbf{v}), \mathbf{u} - \mathbf{v} \rangle]$. For the simplex $\Delta_m$, we choose the entropy function $\nu_q(\mathbf{q}) = \sum_{i=1}^{m} q_i \ln q_i$, which is 1-strongly convex with respect to the vector $\ell_1$-norm $\|\cdot\|_1$, as the distance-generating function. Similarly, $B_q(\cdot, \cdot)$ is the Bregman distance associated with $\nu_q(\cdot)$.

Then, we introduce the standard assumptions about the domain, and the loss function.

**Assumption 1** *The domain $\mathcal{W}$ is convex and its diameter measured by $\nu_w(\cdot)$ is bounded by $D$, i.e.,*

$$\max_{\mathbf{w} \in \mathcal{W}} \nu_w(\mathbf{w}) - \min_{\mathbf{w} \in \mathcal{W}} \nu_w(\mathbf{w}) \leq D^2. \tag{4}$$

For $\Delta_m$, it is easy to verify that its diameter measured by the entropy function is bounded by $\sqrt{\ln m}$.

**Assumption 2** *For all $i \in [m]$, the risk function $R_i(\mathbf{w}) = \mathrm{E}_{\mathbf{z} \sim \mathcal{P}_i}[\ell(\mathbf{w}; \mathbf{z})]$ is convex.*

**Assumption 3** *For all $i \in [m]$, we have*

$$0 \leq \ell(\mathbf{w}; \mathbf{z}) \leq 1, \ \forall \mathbf{w} \in \mathcal{W}, \ \mathbf{z} \sim \mathcal{P}_i. \tag{5}$$

**Assumption 4** *For all $i \in [m]$, we have*

$$\|\nabla \ell(\mathbf{w}; \mathbf{z})\|_{w,*} \leq G, \ \forall \mathbf{w} \in \mathcal{W}, \ \mathbf{z} \sim \mathcal{P}_i \tag{6}$$

*where $\|\cdot\|_{w,*}$ is the dual norm of $\|\cdot\|_w$.*

**Algorithm 1** Stochastic Mirror Descent for GDRO

---
**Input**: Two step sizes: $\eta_w$ and $\eta_q$
1: Initialize $\mathbf{w}_1 = \operatorname{argmin}_{\mathbf{w} \in \mathcal{W}} \nu_w(\mathbf{w})$, and $\mathbf{q}_1 = [1/m, \ldots, 1/m]^\top \in \mathbb{R}^m$
2: **for** $t = 1$ to $T$ **do**
3:     For each $i \in [m]$, draw a sample $\mathbf{z}_t^{(i)}$ from distribution $\mathcal{P}_i$
4:     Construct the stochastic gradients defined in (9)
5:     Update $\mathbf{w}_t$ and $\mathbf{q}_t$ according to (11) and (12), respectively
6: **end for**
7: **return** $\bar{\mathbf{w}} = \frac{1}{T} \sum_{t=1}^T \mathbf{w}_t$ and $\bar{\mathbf{q}} = \frac{1}{T} \sum_{t=1}^T \mathbf{q}_t$

---

Last, we discuss the performance measure. To analyze the convergence property, we measure the quality of an approximate solution $(\bar{\mathbf{w}}, \bar{\mathbf{q}})$ to (3) by the error

$$\epsilon_\phi(\bar{\mathbf{w}}, \bar{\mathbf{q}}) = \max_{\mathbf{q} \in \Delta_m} \phi(\bar{\mathbf{w}}, \mathbf{q}) - \min_{\mathbf{w} \in \mathcal{W}} \phi(\mathbf{w}, \bar{\mathbf{q}}) \tag{7}$$

which directly controls the optimality of $\bar{\mathbf{w}}$ to the original problem (2), since

$$\max_{i \in [m]} R_i(\bar{\mathbf{w}}) - \min_{\mathbf{w} \in \mathcal{W}} \max_{i \in [m]} R_i(\mathbf{w}) \le \max_{\mathbf{q} \in \Delta_m} \sum_{i=1}^m q_i R_i(\bar{\mathbf{w}}) - \min_{\mathbf{w} \in \mathcal{W}} \sum_{i=1}^m \bar{q}_i R_i(\mathbf{w}) = \epsilon_\phi(\bar{\mathbf{w}}, \bar{\mathbf{q}}). \tag{8}$$

## 2.2 Stochastic Mirror Descent for GDRO

To apply SMD, the key is to construct stochastic gradients of the function $\phi(\mathbf{w}, \mathbf{q})$ in (3). In each round $t$, denote by $\mathbf{w}_t$ and $\mathbf{q}_t$ the current solutions. We draw one sample $\mathbf{z}_t^{(i)}$ from every distribution $\mathcal{P}_i$, and define stochastic gradients as

$$\mathbf{g}_w(\mathbf{w}_t, \mathbf{q}_t) = \sum_{i=1}^m q_{t,i} \nabla \ell(\mathbf{w}_t; \mathbf{z}_t^{(i)}), \text{ and } \mathbf{g}_q(\mathbf{w}_t, \mathbf{q}_t) = [\ell(\mathbf{w}_t; \mathbf{z}_t^{(1)}), \ldots, \ell(\mathbf{w}_t; \mathbf{z}_t^{(m)})]^\top. \tag{9}$$

It is worth mentioning that the construction of $\mathbf{g}_w(\mathbf{w}_t, \mathbf{q}_t)$ can be further simplified to

$$\tilde{\mathbf{g}}_w(\mathbf{w}_t, \mathbf{q}_t) = \nabla \ell(\mathbf{w}_t; \mathbf{z}_t^{(i_t)}) \tag{10}$$

where $i_t \in [m]$ is drawn randomly according to the probability $\mathbf{q}_t$.

Then, we utilize SMD to update $\mathbf{w}_t$ and $\mathbf{q}_t$:

$$\mathbf{w}_{t+1} = \operatorname*{argmin}_{\mathbf{w} \in \mathcal{W}} \left\{ \eta_w \langle \mathbf{g}_w(\mathbf{w}_t, \mathbf{q}_t), \mathbf{w} - \mathbf{w}_t \rangle + B_w(\mathbf{w}, \mathbf{w}_t) \right\}, \tag{11}$$

$$\mathbf{q}_{t+1} = \operatorname*{argmin}_{\mathbf{q} \in \Delta_m} \left\{ \eta_q \langle -\mathbf{g}_q(\mathbf{w}_t, \mathbf{q}_t), \mathbf{q} - \mathbf{q}_t \rangle + B_q(\mathbf{q}, \mathbf{q}_t) \right\} \tag{12}$$

where $\eta_w > 0$ and $\eta_q > 0$ are two step sizes that will be determined later. The updating rule of $\mathbf{w}_t$ depends on the choice of the distance-generating function $\nu_w(\cdot)$. Since $B_q(\mathbf{q}, \mathbf{q}_t)$ is defined in terms of the negative entropy, (12) is equivalent to

$$q_{t+1,i} = \frac{q_{t,i} \exp\left(\eta_q \ell(\mathbf{w}_t; \mathbf{z}_t^{(i)})\right)}{\sum_{j=1}^m q_{t,j} \exp\left(\eta_q \ell(\mathbf{w}_t; \mathbf{z}_t^{(j)})\right)}, \ \forall i \in [m] \tag{13}$$

which is the Hedge algorithm [Freund and Schapire, 1997] applied to a maximization problem. In the beginning, we set $\mathbf{w}_1 = \operatorname{argmin}_{\mathbf{w} \in \mathcal{W}} \nu_w(\mathbf{w})$, and $\mathbf{q}_1 = \frac{1}{m} \mathbf{1}_m$, where $\mathbf{1}_m$ is the $m$-dimensional vector consisting of 1's. In the last step, we return the averaged iterates $\bar{\mathbf{w}} = \frac{1}{T} \sum_{t=1}^T \mathbf{w}_t$ and $\bar{\mathbf{q}} = \frac{1}{T} \sum_{t=1}^T \mathbf{q}_t$ as final solutions. The completed procedure is summarized in Algorithm 1.

Based on the theoretical guarantee of SMD [Nemirovski et al., 2009, §3.1], we have the following theorem.

**Theorem 1** *Under Assumptions 1, 2, 3 and 4, and setting* $\eta_w = D^2 \sqrt{\frac{8}{5T(D^2 G^2 + \ln m)}}$ *and* $\eta_q = (\ln m) \sqrt{\frac{8}{5T(D^2 G^2 + \ln m)}}$ *in Algorithm 1, with probability at least* $1 - \delta$, *we have*

$$\epsilon_\phi(\bar{\mathbf{w}}, \bar{\mathbf{q}}) \le \left(8 + 2 \ln \frac{2}{\delta}\right) \sqrt{\frac{10(D^2 G^2 + \ln m)}{T}}.$$

**Remark 1** Theorem 1 shows that Algorithm 1 achieves an $O(\sqrt{(\log m)/T})$ convergence rate. Since it consumes $m$ samples per iteration, the sample complexity is $O(m(\log m)/\epsilon^2)$, which nearly matches the $\Omega(m/\epsilon^2)$ lower bound [Soma et al., 2022, Theorem 5]. Due to space limitations, we defer all the proofs to Appendix B, and omit expectation bounds.

**Comparisons with Sagawa et al. [2020]** In their stochastic algorithm, Sagawa et al. [2020] generate a random index $i_t \in [m]$ uniformly in each round $t$, and draw 1 sample $\mathbf{z}_t^{(i_t)}$ from $\mathcal{P}_{i_t}$. The stochastic gradients are constructed as follows:

$$\hat{\mathbf{g}}_w(\mathbf{w}_t, \mathbf{q}_t) = q_{t,i_t} m \nabla \ell(\mathbf{w}_t; \mathbf{z}_t^{(i_t)}), \text{ and } \hat{\mathbf{g}}_q(\mathbf{w}_t, \mathbf{q}_t) = [0, \ldots, m\ell(\mathbf{w}_t; \mathbf{z}_t^{(i_t)}), \ldots, 0]^\top \quad (14)$$

where $\hat{\mathbf{g}}_q(\mathbf{w}_t, \mathbf{q}_t)$ is a vector with $m\ell(\mathbf{w}_t; \mathbf{z}_t^{(i_t)})$ in position $i_t$ and 0 elsewhere. Then, the two stochastic gradients are used to update $\mathbf{w}_t$ and $\mathbf{q}_t$, in the same way as (11) and (12). However, it only attains a slow convergence rate: $O(m\sqrt{(\log m)/T})$, leading to an $O(m^2(\log m)/\epsilon^2)$ sample complexity, which is higher than that of Algorithm 1 by a factor of $m$. The slow convergence is due to the fact that the optimization error depends on the dual norm of the stochastic gradients in (14), which blows up by a factor of $m$, compared with the gradients in (9).

**Comparisons with Haghtalab et al. [2022]** To reduce the number of samples required in each round, Haghtalab et al. [2022] propose to reuse samples for multiple iterations. To approximate $\nabla_{\mathbf{w}}\phi(\mathbf{w}_t, \mathbf{q}_t)$, they construct the stochastic gradient $\tilde{\mathbf{g}}_w(\mathbf{w}_t, \mathbf{q}_t)$ in the same way as (10), which needs 1 sample. To approximate $\nabla_{\mathbf{q}}\phi(\mathbf{w}_t, \mathbf{q}_t)$, they draw $m$ samples $\mathbf{z}_\tau^{(1)}, \ldots, \mathbf{z}_\tau^{(m)}$, one from each distribution, at round $\tau = mk + 1$, $k \in \mathbb{Z}$, and reuse them for $m$ iterations to construct

$$\mathbf{g}_q'(\mathbf{w}_t, \mathbf{q}_t) = [\ell(\mathbf{w}_t; \mathbf{z}_\tau^{(1)}), \ldots, \ell(\mathbf{w}_t; \mathbf{z}_\tau^{(m)})]^\top, \ t = \tau, \ldots, \tau + m - 1. \quad (15)$$

Then, they treat $\tilde{\mathbf{g}}_w(\mathbf{w}_t, \mathbf{q}_t)$ and $\mathbf{g}_q'(\mathbf{w}_t, \mathbf{q}_t)$ as stochastic gradients, and update $\mathbf{w}_t$ and $\mathbf{q}_t$ by SMD. In this way, their algorithm uses 2 samples on average in each iteration. However, the gradient in (15) is no longer an unbiased estimator of the true gradient $\mathbf{g}_q(\mathbf{w}_t, \mathbf{q}_t)$ at rounds $t = \tau + 2, \ldots, \tau + m - 1$, making their analysis ungrounded. To see this, from the updating rule of SMD, we know that $\mathbf{w}_{\tau+2}$ depends on $\mathbf{q}_{\tau+1}$, which in turn depends on the $m$ samples drawn at round $\tau$, and thus

$$\mathrm{E}\left[\ell(\mathbf{w}_{\tau+2}; \mathbf{z}_\tau^{(i)})\right] \neq R_i(\mathbf{w}_{\tau+2}), \ i = 1, \ldots, m.$$

**Remark 2** As shown in (10), we can use 1 sample to construct a stochastic gradient for $\mathbf{w}_t$ with small norm, since $\|\tilde{\mathbf{g}}_w(\mathbf{w}_t, \mathbf{q}_t)\|_{w,*} \leq G$ under Assumption 4. Thus, it is relatively easy to control the error related to $\mathbf{w}_t$. However, we do not have such guarantees for the stochastic gradient of $\mathbf{q}_t$. Recall that the infinity norm of $\hat{\mathbf{g}}_q(\mathbf{w}_t, \mathbf{q}_t)$ in (14) is upper bounded by $m$. The reason is that we insist on the unbiasedness of the stochastic gradient, which leads to a large variance. In the next section, we borrow techniques from online learning to better balance the bias and the variance.

## 2.3 Non-oblivious Online Learning for GDRO

In the studies of convex-concave saddle-point problems, it is now well-known that they can be solved by playing two online learning algorithms against each other [Freund and Schapire, 1999, Rakhlin and Sridharan, 2013, Syrgkanis et al., 2015]. With this transformation, we can exploit no-regret algorithms developed in online learning to bound the optimization error. To solve problem (3), we ask the 1st player to minimize a sequence of convex functions

$$\phi(\mathbf{w}, \mathbf{q}_1) = \sum_{i=1}^m q_{1,i} R_i(\mathbf{w}), \ \ \phi(\mathbf{w}, \mathbf{q}_2) = \sum_{i=1}^m q_{2,i} R_i(\mathbf{w}), \ \cdots, \ \phi(\mathbf{w}, \mathbf{q}_T) = \sum_{i=1}^m q_{T,i} R_i(\mathbf{w})$$

subject to the constraint $\mathbf{w} \in \mathcal{W}$, and the 2nd player to maximize a sequence of linear functions

$$\phi(\mathbf{w}_1, \mathbf{q}) = \sum_{i=1}^m q_i R_i(\mathbf{w}_1), \ \ \phi(\mathbf{w}_2, \mathbf{q}) = \sum_{i=1}^m q_i R_i(\mathbf{w}_2), \ \cdots, \ \phi(\mathbf{w}_T, \mathbf{q}) = \sum_{i=1}^m q_i R_i(\mathbf{w}_T)$$

subject to the constraint $\mathbf{q} \in \Delta_m$. We highlight that there exists an important difference between our stochastic convex-concave problem and its deterministic counterpart. Here, the two players cannot

---

**Algorithm 2** Non-oblivious Online Learning for GDRO

---

**Input**: Two step sizes: $\eta_w$ and $\eta_q$; IX coefficient $\gamma$

1: Initialize $\mathbf{w}_1 = \operatorname{argmin}_{\mathbf{w} \in \mathcal{W}} \nu_w(\mathbf{w})$, and $\mathbf{q}_1 = [1/m, \ldots, 1/m]^\top \in \mathbb{R}^m$
2: **for** $t = 1$ to $T$ **do**
2:     Generate $i_t \in [m]$ according to $\mathbf{q}_t$, and then draw a sample $\mathbf{z}_t^{(i_t)}$ from $\mathbf{q}_{i_t}$
3:     Construct the stochastic gradient in (10) and the IX loss estimator in (17)
4:     Update $\mathbf{w}_t$ and $\mathbf{q}_t$ according to (16) and (18), respectively
5: **end for**
6: **return** $\bar{\mathbf{w}} = \frac{1}{T} \sum_{t=1}^T \mathbf{w}_t$ and $\bar{\mathbf{q}} = \frac{1}{T} \sum_{t=1}^T \mathbf{q}_t$

---

directly observe the loss function, and can only approximate $R_i(\mathbf{w}) = \mathrm{E}_{\mathbf{z} \sim \mathcal{P}_i}[\ell(\mathbf{w}; \mathbf{z})]$ by drawing samples from $\mathcal{P}_i$. The stochastic setting makes the problem more challenging, and in particular, we need to take care of the *non-oblivious* nature of the learning process. Here, "non-oblivious" refers to the fact that the online functions depend on the past decisions of the learner.

Next, we discuss the online algorithms that will be used by the two players. As shown in Section 2.2, the 1st player can easily obtain a stochastic gradient with small norm by using 1 sample. So, we model the problem faced by the 1st player as "non-oblivious online convex optimization (OCO) with stochastic gradients", and still use SMD to update its solution. In each round $t$, with 1 sample drawn from $\mathcal{P}_i$, the 2nd player can estimate the value of $R_i(\mathbf{w}_t)$ which is the coefficient of $q_i$. Since the 2nd player is maximizing a linear function over the simplex, the problem can be modeled as "non-oblivious multi-armed bandits (MAB) with stochastic rewards". And fortunately, we have powerful online algorithms for non-oblivious MAB [Auer et al., 2002, Lattimore and Szepesvári, 2020], whose regret has a sublinear dependence on $m$. In this paper, we choose the Exp3-IX algorithm [Neu, 2015], and generalize its theoretical guarantee to stochastic rewards.

The complete procedure is presented in Algorithm 2, and we explain key steps below. In each round $t$, we generate an index $i_t \in [m]$ from the probability distribution $\mathbf{q}_t$, and then draw a sample $\mathbf{z}_t^{(i_t)}$ from the distribution $\mathbf{q}_{i_t}$. With the stochastic gradient in (10), we use SMD to update $\mathbf{w}_t$:

$$\mathbf{w}_{t+1} = \operatorname*{argmin}_{\mathbf{w} \in \mathcal{W}} \left\{ \eta_w \langle \tilde{\mathbf{g}}_w(\mathbf{w}_t, \mathbf{q}_t), \mathbf{w} - \mathbf{w}_t \rangle + B_w(\mathbf{w}, \mathbf{w}_t) \right\} \tag{16}$$

Then, we reuse the sample $\mathbf{z}_t^{(i_t)}$ to update $\mathbf{q}_t$ according to Exp3-IX, which first constructs the Implicit-eXploration (IX) loss estimator [Kocák et al., 2014]:

$$\tilde{s}_{t,i} = \frac{1 - \ell(\mathbf{w}_t, \mathbf{z}_t^{(i_t)})}{q_{t,i} + \gamma} \cdot \mathbb{I}[i_t = i], \ \forall i \in [m], \tag{17}$$

where $\gamma > 0$ is the IX coefficient and $\mathbb{I}[A]$ equals to 1 when the event $A$ is true and 0 otherwise, and then performs a mirror descent update:

$$\mathbf{q}_{t+1} = \operatorname*{argmin}_{\mathbf{q} \in \Delta_m} \left\{ \eta_q \langle \tilde{\mathbf{s}}_t, \mathbf{q} - \mathbf{q}_t \rangle + B_q(\mathbf{q}, \mathbf{q}_t) \right\}. \tag{18}$$

We present the theoretical guarantee of Algorithm 2.

**Theorem 2** *Under Assumptions 1, 2, 3 and 4, and setting $\eta_w = \frac{2D}{G\sqrt{5T}}$, $\eta_q = \sqrt{\frac{\ln m}{mT}}$ and $\gamma = \frac{\eta_q}{2}$ in Algorithm 2, with probability at least $1 - \delta$, we have*

$$\epsilon_\phi(\bar{\mathbf{w}}, \bar{\mathbf{q}})$$

$$\leq DG\sqrt{\frac{1}{T}} \left( 2\sqrt{5} + 8\sqrt{\ln \frac{2}{\delta}} \right) + 3\sqrt{\frac{m \ln m}{T}} + \sqrt{\frac{1}{2T}} + \left( \sqrt{\frac{m}{T \ln m}} + \sqrt{\frac{1}{2T}} + \frac{1}{T} \right) \ln \frac{6}{\delta}. \tag{19}$$

**Remark 3** The above theorem shows that with 1 sample per iteration, Algorithm 2 is able to achieve an $O(\sqrt{m(\log m)/T})$ convergence rate, thus maintaining the $O(m(\log m)/\epsilon^2)$ sample complexity.

**Comparisons with Soma et al. [2022]** In a recent work, Soma et al. [2022] have deployed online algorithms to optimize $\mathbf{w}$ and $\mathbf{q}$, but did not consider the non-oblivious property. As a result, their

theoretical guarantees, which build upon the analysis for oblivious online learning [Orabona, 2019], cannot justify the optimality of their algorithm for (3). Specifically, their results imply that for any *fixed* $\mathbf{w}$ and $\mathbf{q}$ that are independent from $\bar{\mathbf{w}}$ and $\bar{\mathbf{q}}$ [Soma et al., 2022, Theorem 3],

$$\mathrm{E}\left[\phi(\bar{\mathbf{w}}, \mathbf{q}) - \phi(\mathbf{w}, \bar{\mathbf{q}})\right] = O\left(\sqrt{\frac{m}{T}}\right). \tag{20}$$

However, (20) cannot be used to bound $\epsilon_\phi(\bar{\mathbf{w}}, \bar{\mathbf{q}})$ in (7), because of the dependency issue. To be more clear, we have $\epsilon_\phi(\bar{\mathbf{w}}, \bar{\mathbf{q}}) = \max_{\mathbf{q} \in \Delta_m} \phi(\bar{\mathbf{w}}, \mathbf{q}) - \min_{\mathbf{w} \in \mathcal{W}} \phi(\mathbf{w}, \bar{\mathbf{q}}) = \phi(\bar{\mathbf{w}}, \widehat{\mathbf{q}}) - \phi(\widehat{\mathbf{w}}, \bar{\mathbf{q}})$, where $\widehat{\mathbf{w}} = \operatorname{argmin}_{\mathbf{w} \in \mathcal{W}} \phi(\mathbf{w}, \bar{\mathbf{q}})$ and $\widehat{\mathbf{q}} = \operatorname{argmax}_{\mathbf{q} \in \Delta_m} \phi(\bar{\mathbf{w}}, \mathbf{q})$ *depend* on $\bar{\mathbf{w}}$ and $\bar{\mathbf{q}}$.

**Remark 4** After we pointed out the issue of reusing samples, Haghtalab et al. [2023] modified their method by incorporating bandits algorithms to optimize $\mathbf{q}$. From our understanding, the idea of applying bandits to GDRO is *firstly* proposed by Soma et al. [2022], and subsequently refined by us.

## 3  Weighted GDRO and SA Approaches

When designing SA approaches for GDRO, it is common to assume that the algorithms are free to draw samples from every distribution [Sagawa et al., 2020], as we do in Section 2. However, this assumption may not hold in practice. For example, data collection costs can vary widely among distributions [Radivojac et al., 2004], and data collected from various channels can have different throughputs [Zhou, 2023]. In this section, we investigate the scenario where the number of samples can be drawn from each distribution could be different. Denote by $n_i$ the number of samples that can be drawn from $\mathcal{P}_i$. Without loss of generality, we assume that $n_1 \geq n_2 \geq \cdots \geq n_m$. Note that we have a straightforward **Baseline** which just runs Algorithm 1 for $n_m$ iterations, and the optimization error $\epsilon_\phi(\bar{\mathbf{w}}, \bar{\mathbf{q}}) = O(\sqrt{(\log m)/n_m})$.

### 3.1  Stochastic Mirror Descent with Non-uniform Sampling

To meet the budget, we propose to incorporate non-uniform sampling into SMD. Specifically, in round $t$, we first generate a set of Bernoulli random variables $\{b_t^{(1)}, \ldots, b_t^{(m)}\}$ with $\Pr[b_t^{(i)} = 1] = p_i$ to determine whether to sample from each distribution. If $b_t^{(i)} = 1$, we draw a sample $\mathbf{z}_t^{(i)}$ from $\mathcal{P}_i$. Now, the question is how to construct stochastic gradients from those samples. Let $\mathcal{C}_t = \{i | b_t^{(i)} = 1\}$ be the indices of selected distributions. If we stick to the original problem in (3), then the stochastic gradients should be constructed in the following way

$$\mathbf{g}_w(\mathbf{w}_t, \mathbf{q}_t) = \sum_{i \in C_t} \frac{q_{t,i}}{p_i} \nabla \ell(\mathbf{w}_t; \mathbf{z}_t^{(i)}), \text{ and } [\mathbf{g}_q(\mathbf{w}_t, \mathbf{q}_t)]_i = \begin{cases} \ell(\mathbf{w}_t; \mathbf{z}_t^{(i)})/p_i, & i \in \mathcal{C}_t \\ 0, & \text{otherwise} \end{cases} \tag{21}$$

to ensure unbiasedness. Then, they can be used by SMD to update $\mathbf{w}_t$ and $\mathbf{q}_t$. After $n_1$ iterations, the *expected* number of samples drawn from $\mathcal{P}_i$ will be $n_1 p_i = n_i$, and thus the budget is satisfied in expectation. To analyze the optimization error, we need to bound the norm of stochastic gradients in (21). To this end, we have $\|\mathbf{g}_w(\mathbf{w}_t, \mathbf{q}_t)\|_{w,*} \leq G n_1/n_m$ and $\|\mathbf{g}_q(\mathbf{w}_t, \mathbf{q}_t)_i\|_\infty \leq n_1/n_m$. Following the arguments of Theorem 1, we can prove that the error $\epsilon_\phi(\bar{\mathbf{w}}, \bar{\mathbf{q}}) = O(\sqrt{(\log m)/n_1} \cdot n_1/n_m) = O(\sqrt{n_1 \log m}/n_m)$, which is even larger than the $O(\sqrt{(\log m)/n_m})$ error of the Baseline.

In the following, we demonstrate that a simple twist of the above procedure can still yield meaningful results that are complementary to the Baseline. We observe that the large norm of the stochastic gradients in (21) is caused by the inverse probability $1/p_i$. A natural idea is to ignore $1/p_i$, and define the following stochastic gradients:

$$\mathbf{g}_w(\mathbf{w}_t, \mathbf{q}_t) = \sum_{i \in C_t} q_{t,i} \nabla \ell(\mathbf{w}_t; \mathbf{z}_t^{(i)}), \text{ and } [\mathbf{g}_q(\mathbf{w}_t, \mathbf{q}_t)]_i = \begin{cases} \ell(\mathbf{w}_t; \mathbf{z}_t^{(i)}), & i \in \mathcal{C}_t \\ 0, & \text{otherwise.} \end{cases} \tag{22}$$

In this way, they are no longer stochastic gradients of (3), but can be treated as stochastic gradients of a weighted GDRO problem:

$$\min_{\mathbf{w} \in \mathcal{W}} \max_{\mathbf{q} \in \Delta_m} \left\{\varphi(\mathbf{w}, \mathbf{q}) = \sum_{i=1}^m q_i p_i R_i(\mathbf{w})\right\} \tag{23}$$

---

**Algorithm 3** Stochastic Mirror Descent for Weighted GDRO

---

**Input**: Two step sizes: $\eta_w$ and $\eta_q$

1: Initialize $\mathbf{w}_1 = \operatorname{argmin}_{\mathbf{w} \in \mathcal{W}} \nu_w(\mathbf{w})$, and $\mathbf{q}_1 = [1/m, \ldots, 1/m]^\top \in \mathbb{R}^m$
2: **for** $t = 1$ to $n_1$ **do**
3:     For each $i \in [m]$, generate a Bernoulli random variable $b_t^{(i)}$ with $\Pr[b_t^{(i)} = 1] = p_i$, and if $b_t^{(i)} = 1$, draw a sample $\mathbf{z}_t^{(i)}$ from distribution $\mathcal{P}_i$
4:     Construct the stochastic gradients defined in (22)
5:     Update $\mathbf{w}_t$ and $\mathbf{q}_t$ according to (11) and (12), respectively
6: **end for**
7: **return** $\bar{\mathbf{w}} = \frac{1}{n_1} \sum_{t=1}^{n_1} \mathbf{w}_t$ and $\bar{\mathbf{q}} = \frac{1}{n_1} \sum_{t=1}^{n_1} \mathbf{q}_t$

---

where each risk $R_i(\cdot)$ is scaled by a factor $p_i$. Based on the gradients in (22), we still use (11) and (12) to update $\mathbf{w}_t$ and $\mathbf{q}_t$. We summarize the complete procedure in Algorithm 3.

We omit the optimization error of Algorithm 3 for (23), since it has exactly the same form as Theorem 1. What we are really interested in is the theoretical guarantee of its solution on multiple distributions. To this end, we have the following theorem.

**Theorem 3** *Under Assumptions 1, 2, 3 and 4, and setting $\eta_w = D^2 \sqrt{\frac{8}{5n_1(D^2G^2 + \ln m)}}$ and $\eta_q = (\ln m)\sqrt{\frac{8}{5n_1(D^2G^2 + \ln m)}}$ in Algorithm 3, with probability at least $1 - \delta$, we have*

$$R_i(\bar{\mathbf{w}}) - \frac{n_1}{n_i} p_\varphi^* \leq \frac{1}{p_i} \mu(\delta) \sqrt{\frac{10(D^2G^2 + \ln m)}{n_1}} = \mu(\delta) \frac{\sqrt{10(D^2G^2 + \ln m)n_1}}{n_i}, \ \forall i \in [m]$$

*where $p_\varphi^*$ is the optimal value of (23) and $\mu(\delta) = 8 + 2\ln\frac{2}{\delta}$.*

**Remark 5** We see that Algorithm 3 exhibits a *distribution-dependent* convergence behavior: The larger the number of samples $n_i$, the smaller the target risk $n_1 p_\varphi^*/n_i$, and the faster the convergence rate $O(\sqrt{n_1 \log m}/n_i)$. Note that its rate is always better than the $O(\sqrt{n_1 \log m}/n_m)$ rate of SMD with (21) as gradients. Furthermore, it converges faster than the Baseline when $n_i \geq \sqrt{n_1 n_m}$. In particular, for distribution $\mathcal{P}_1$, Algorithm 3 attains an $O(\sqrt{(\log m)/n_1})$ rate, which almost matches the optimal $O(\sqrt{1/n_1})$ rate of learning from a single distribution. Finally, we would like to emphasize that a similar idea of introducing "scale factors" has been used by Juditsky et al. [2011, §4.3.1] for stochastic semidefinite feasibility problems and Agarwal and Zhang [2022] for empirical MRO.

### 3.2 Stochastic Mirror-Prox Algorithm with Mini-batches

In Algorithm 3, distributions with more samples take their advantage by appearing more frequently in the stochastic gradients. In this section, we propose a different way, which lets them reduce the variance in the elements of stochastic gradients by mini-batches [Roux et al., 2008].

The basic idea is as follows. We run our algorithm for a small number of iterations $\bar{n}$ that is no larger than $n_m$. Then, in each iteration, we draw a mini-batch of $n_i/\bar{n}$ samples from every distribution $\mathcal{P}_i$. For $\mathcal{P}_i$ with more samples, we can estimate the associated risk $R_i(\cdot)$ and its gradient more accurately, i.e., with a smaller variance. However, to make this idea work, we need to tackle two obstacles: (i) the performance of the SA algorithm should depend on the variance of gradients instead of the norm, and for this reason SMD is unsuitable; (ii) even some elements of the stochastic gradient have small variances, the entire gradient may still have a large variance. To address the first challenge, we resort to a more advanced SA approach—stochastic mirror-prox algorithm (SMPA), whose convergence rate depends on the variance [Juditsky et al., 2011]. To overcome the second challenge, we again introduce scale factors into the optimization problem and the stochastic gradients.

In SMPA, we need to maintain two sets of solutions: $(\mathbf{w}_t, \mathbf{q}_t)$ and $(\mathbf{w}_t', \mathbf{q}_t')$. In each round $t$, we first draw $n_i/n_m$ samples from every distribution $\mathcal{P}_i$, denoted by $\mathbf{z}_t^{(i,j)}$, $j = 1, \ldots, n_i/n_m$. Then, we use them to construct stochastic gradients at $(\mathbf{w}_t', \mathbf{q}_t')$ of a weighted GDRO problem (23), where the

---
**Algorithm 4** Stochastic Mirror-Prox Algorithm for Weighted GDRO
---
**Input**: Two step sizes: $\eta_w$ and $\eta_q$

1: Initialize $\mathbf{w}_1' = \operatorname{argmin}_{\mathbf{w} \in \mathcal{W}} \nu_w(\mathbf{w})$, and $\mathbf{q}_1' = [1/m, \ldots, 1/m]^\top \in \mathbb{R}^m$
2: **for** $t = 1$ to $n_m/2$ **do**
3:    For each $i \in [m]$, draw $n_i/n_m$ samples $\{\mathbf{z}_t^{(i,j)} : j = 1, \ldots, n_i/n_m\}$ from distribution $\mathcal{P}_i$
4:    Construct the stochastic gradients defined in (24)
5:    Calculate $\mathbf{w}_{t+1}$ and $\mathbf{q}_{t+1}$ according to (25) and (26), respectively
6:    For each $i \in [m]$, draw $n_i/n_m$ samples $\{\hat{\mathbf{z}}_t^{(i,j)} : j = 1, \ldots, n_i/n_m\}$ from distribution $\mathcal{P}_i$
7:    Construct the stochastic gradients defined in (27)
8:    Calculate $\mathbf{w}_{t+1}'$ and $\mathbf{q}_{t+1}'$ according to (28) and (29), respectively
9: **end for**
10: **return** $\bar{\mathbf{w}} = \frac{2}{n_m} \sum_{t=2}^{1+n_m/2} \mathbf{w}_t$ and $\bar{\mathbf{q}} = \frac{2}{n_m} \sum_{t=2}^{1+n_m/2} \mathbf{q}_t$
---

value of $p_i$ will be determined later. Specifically, we define

$$
\mathbf{g}_w(\mathbf{w}_t', \mathbf{q}_t') = \sum_{i=1}^m q_{t,i}' p_i \left( \frac{n_m}{n_i} \sum_{j=1}^{n_i/n_m} \nabla \ell(\mathbf{w}_t'; \mathbf{z}_t^{(i,j)}) \right),
$$

$$
\mathbf{g}_q(\mathbf{w}_t', \mathbf{q}_t') = \left[ p_1 \frac{n_m}{n_1} \sum_{j=1}^{n_1/n_m} \ell(\mathbf{w}_t'; \mathbf{z}_t^{(1,j)}), \ldots, p_m \ell(\mathbf{w}_t'; \mathbf{z}_t^{(m)}) \right]^\top. \tag{24}
$$

Let's take the stochastic gradient $\mathbf{g}_q(\mathbf{w}_t', \mathbf{q}_t')$, whose variance will be measured in terms of $\| \cdot \|_\infty$, as an example to explain the intuition of inserting $p_i$. Define $u_i = \frac{n_m}{n_i} \sum_{j=1}^{n_i/n_m} \ell(\mathbf{w}_t'; \mathbf{z}_t^{(i,j)})$. With a larger mini-batch size $n_i/n_m$, $u_i$ will approximate $R_i(\mathbf{w}_t')$ more accurately, and thus have a smaller variance. Then, it allows us to insert a larger value of $p_i$, without affecting the variance of $\|\mathbf{g}_q(\mathbf{w}_t', \mathbf{q}_t')\|_\infty$, since $\| \cdot \|_\infty$ is insensitive to perturbations of small elements. Similar to the case in Theorem 3, the convergence rate of $R_i(\cdot)$ depends on $1/p_i$, and becomes faster if $p_i$ is larger.

Based on (24), we use SMD to update $(\mathbf{w}_t', \mathbf{q}_t')$, and denote the solution by $(\mathbf{w}_{t+1}, \mathbf{q}_{t+1})$:

$$
\mathbf{w}_{t+1} = \operatorname*{argmin}_{\mathbf{w} \in \mathcal{W}} \left\{ \eta_w \langle \mathbf{g}_w(\mathbf{w}_t', \mathbf{q}_t'), \mathbf{w} - \mathbf{w}_t' \rangle + B_w(\mathbf{w}, \mathbf{w}_t') \right\}, \tag{25}
$$

$$
\mathbf{q}_{t+1} = \operatorname*{argmin}_{\mathbf{q} \in \Delta_m} \left\{ \eta_q \langle -\mathbf{g}_q(\mathbf{w}_t', \mathbf{q}_t'), \mathbf{q} - \mathbf{q}_t' \rangle + B_q(\mathbf{q}, \mathbf{q}_t') \right\}. \tag{26}
$$

Next, we draw another $n_i/n_m$ samples from each distribution $\mathcal{P}_i$, denoted by $\hat{\mathbf{z}}_t^{(i,j)}$, $j = 1, \ldots, n_i/n_m$, to construct stochastic gradients at $(\mathbf{w}_{t+1}, \mathbf{q}_{t+1})$:

$$
\mathbf{g}_w(\mathbf{w}_{t+1}, \mathbf{q}_{t+1}) = \sum_{i=1}^m q_{t+1,i} p_i \left( \frac{n_m}{n_i} \sum_{j=1}^{n_i/n_m} \nabla \ell(\mathbf{w}_{t+1}; \hat{\mathbf{z}}_t^{(i,j)}) \right),
$$

$$
\mathbf{g}_q(\mathbf{w}_{t+1}, \mathbf{q}_{t+1}) = \left[ p_1 \frac{n_m}{n_1} \sum_{j=1}^{n_1/n_m} \ell(\mathbf{w}_{t+1}; \hat{\mathbf{z}}_t^{(1,j)}), \ldots, p_m \ell(\mathbf{w}_{t+1}; \hat{\mathbf{z}}_t^{(m)}) \right]^\top. \tag{27}
$$

Then, we use them to update $(\mathbf{w}_t', \mathbf{q}_t')$ again, and denote the result by $(\mathbf{w}_{t+1}', \mathbf{q}_{t+1}')$:

$$
\mathbf{w}_{t+1}' = \operatorname*{argmin}_{\mathbf{w} \in \mathcal{W}} \left\{ \eta_w \langle \mathbf{g}_w(\mathbf{w}_{t+1}, \mathbf{q}_{t+1}), \mathbf{w} - \mathbf{w}_t' \rangle + B_w(\mathbf{w}, \mathbf{w}_t') \right\}, \tag{28}
$$

$$
\mathbf{q}_{t+1}' = \operatorname*{argmin}_{\mathbf{q} \in \Delta_m} \left\{ \eta_q \langle -\mathbf{g}_q(\mathbf{w}_{t+1}, \mathbf{q}_{t+1}), \mathbf{q} - \mathbf{q}_t' \rangle + B_q(\mathbf{q}, \mathbf{q}_t') \right\}. \tag{29}
$$

To meet the budget constraints, we repeat the above process for $n_m/2$ iterations. Finally, we return $\bar{\mathbf{w}} = \frac{2}{n_m} \sum_{t=2}^{1+n_m/2} \mathbf{w}_t$ and $\bar{\mathbf{q}} = \frac{2}{n_m} \sum_{t=2}^{1+n_m/2} \mathbf{q}_t$ as solutions. The completed procedure is summarized in Algorithm 4.

To analysis the performance of Algorithm 4, we further assume the risk function $R_i(\cdot)$ is smooth, and the dual norm $\| \cdot \|_{w,*}$ satisfies a regularity condition.

**Assumption 5** *All the risk functions are L-smooth, i.e.,*

$$\|\nabla R_i(\mathbf{w}) - \nabla R_i(\mathbf{w}')\|_{w,*} \leq L\|\mathbf{w} - \mathbf{w}'\|_w, \ \forall \mathbf{w}, \mathbf{w}' \in \mathcal{W}, i \in [m]. \tag{30}$$

Note that even in the studies of stochastic convex optimization (SCO), smoothness is necessary to obtain a variance-based convergence rate [Lan, 2012].

**Assumption 6** *The dual norm $\| \cdot \|_{w,*}$ is $\kappa$-regular for some small constant $\kappa \geq 1$.*

The regularity condition is used when analyzing the effect of mini-batches on stochastic gradients. For a formal definition, please refer to Juditsky and Nemirovski [2008]. Assumption 6 is satisfied by most of papular norms considered in the literature, such as the vector $\ell_p$-norm and the infinity norm.

Then, we have the following theorem for Algorithm 4.

**Theorem 4** *Define*

$$p_{\max} = \max_{i \in [m]} p_i, \quad \omega_{\max} = \max_{i \in [m]} \frac{p_i^2 n_m}{n_i}, \tag{31}$$
$$\widetilde{L} = 2\sqrt{2}p_{\max}(D^2 L + D^2 G\sqrt{\ln m}), \ and \ \sigma^2 = 2c\omega_{\max}(\kappa D^2 G^2 + \ln^2 m)$$

*where $c > 0$ is an absolute constant. Under Assumptions 1, 2, 3, 4, 5 and 6, and setting*

$$\eta_w = 2D^2 \min\left(\frac{1}{\sqrt{3}\widetilde{L}}, \frac{2}{\sqrt{7\sigma^2 n_m}}\right), \ and \ \eta_q = 2\min\left(\frac{1}{\sqrt{3}\widetilde{L}}, \frac{2}{\sqrt{7\sigma^2 n_m}}\right) \ln m$$

*in Algorithm 4, with probability at least $1 - \delta$, we have*

$$R_i(\bar{\mathbf{w}}) - \frac{1}{p_i}p_\varphi^* = \frac{1}{p_i}\left(\frac{7\widetilde{L}}{n_m} + \sqrt{\frac{\sigma^2}{n_m}}\left(14\sqrt{\frac{2}{3}} + 7\sqrt{3\log\frac{2}{\delta}} + \frac{14}{n_m}\log\frac{2}{\delta}\right)\right)$$

*where $p_\varphi^*$ is the optimal value of (23).*
*Furthermore, by setting $p_i$ as*

$$p_i = \frac{1/\sqrt{n_m} + 1}{1/\sqrt{n_m} + \sqrt{n_m/n_i}}, \tag{32}$$

*with high probability, we have*

$$R_i(\bar{\mathbf{w}}) - \frac{1}{p_i}p_\varphi^* = O\left(\left(\frac{1}{n_m} + \frac{1}{\sqrt{n_i}}\right)\sqrt{\kappa + \ln^2 m}\right).$$

**Remark 6** Compared with Algorithm 3, Algorithm 4 has two advantages: (i) the budget constraint is satisfied exactly; (ii) we obtain a faster $O((\log m)/\sqrt{n_i})$ rate for all distributions $\mathcal{P}_i$ such that $n_i \leq n_m^2$, which is much better than the $O(\sqrt{n_1 \ln m}/n_i)$ rate of Algorithm 3, and the $O(\sqrt{(\log m)/n_m})$ rate of the Baseline. For distributions with a larger number of budget, i.e., $n_i > n_m^2$, it maintains a fast $O((\log m)/n_m)$ rate. Since it only updates $n_m$ times, and the best we can expect is the $O(1/n_m)$ rate of deterministic settings [Nemirovski, 2004]. So, there is a performance limit for mini-batch based methods, and after that increasing the batch-size cannot reduce the rate, which consists with the usage of mini-batches in SCO [Cotter et al., 2011, Zhang et al., 2013].

## 4 Conclusion

For the GDRO problem, we develop two SA approaches based on SMD and non-oblivious MAB. Both of them attain the nearly optimal $O(m(\log m)/\epsilon^2)$ sample complexity, but with different number of samples used in each round, which are $m$ and $1$ respectively. Then, we formulate a weighted GDRO problem to handle the scenario in which different distributions have different sample budgets. We first incorporate non-uniform sampling into SMD to satisfy the sample budget in expectation, and deliver distribution-dependent convergence rates. Then, we propose to use mini-batches to meet the budget exactly, deploy SMPA to exploit the small variances, and establish nearly optimal rates for multiple distributions. We have conducted experiments to evaluate our proposed algorithms. The empirical results are presented in Appendix C, and align closely with our theories.

## Acknowledgments and Disclosure of Funding

This work was partially supported by the National Key R&D Program of China (2022ZD0114801), NSFC (62122037, 61921006), National Postdoctoral Program for Innovative Talent, China Postdoctoral Science Foundation (2023M731597), and the major key project of PCL (PCL2021A12).

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

# A  Related Work

Distributionally robust optimization (DRO) stems from the pioneering work of Scarf [1958], and has gained a lot of interest with the advancement of robust optimization [Ben-Tal et al., 2009, 2015]. It has been successfully applied to a variety of machine learning tasks, including adversarial training [Sinha et al., 2018], algorithmic fairness [Hashimoto et al., 2018], class imbalance [Xu et al., 2020], long-tail learning [Samuel and Chechik, 2021], label shift [Zhang et al., 2021], etc.

In general, DRO is formulated to reflect our uncertainty about the target distribution. To ensure good performance under distribution perturbations, it minimizes the risk w.r.t. the worst distribution in an uncertainty set, i.e.,

$$\min_{\mathbf{w} \in \mathcal{W}} \sup_{\mathcal{P} \in \mathcal{S}(\mathcal{P}_0)} \left\{ \mathrm{E}_{\mathbf{z} \sim \mathcal{P}} \left[ \ell(\mathbf{w}; \mathbf{z}) \right] \right\} \tag{33}$$

where $\mathcal{S}(\mathcal{P}_0)$ denotes a set of probability distributions around $\mathcal{P}_0$. In the literature, there mainly exist three ways to construct $\mathcal{S}(\mathcal{P}_0)$: (i) enforcing moment constraints [Delage and Ye, 2010], (ii) defining a neighborhood around $\mathcal{P}_0$ by a distance function such as the $f$-divergence [Ben-Tal et al., 2013], the Wasserstein distance [Kuhn et al., 2019], and the Sinkhorn distance [Wang et al., 2021], and (iii) hypothesis testing of goodness-of-fit [Bertsimas et al., 2018].

By drawing a set of samples from $\mathcal{P}_0$, we can also define an empirical DRO problem, which can be regarded as an SAA approach for solving (33). When the uncertainty set is defined in terms of the Cressie–Read family of $f$-divergences, Duchi and Namkoong [2021] have studied finite sample and asymptotic properties of the empirical solution. Besides, it has been proved that empirical DRO can also benefit the risk minimization problem in (1). Namkoong and Duchi [2017] show that empirical DRO with the $\chi^2$-divergence has the effect of variance regularization, leading to better generalization w.r.t. distribution $\mathcal{P}_0$. Later, Duchi et al. [2021] demonstrate similar behaviors for the $f$-divergence constrained neighborhood, and provide one- and two-sided confidence intervals for the minimal risk in (1). Based on the Wasserstein distance, Esfahani and Kuhn [2018] establish an upper confidence bound on the risk of the empirical solution.

Since (33) is more complex than (1), considerable research efforts were devoted to develop efficient algorithms for DRO and its empirical version. For $\mathcal{P}_0$ with finite support, Ben-Tal et al. [2013, Corollary 3] have demonstrated that (33) with $f$-divergences is equivalent to a convex optimization problem, provided that the loss $\ell(\mathbf{w}; \mathbf{z})$ is convex in $\mathbf{w}$. Actually, this conclusion is true even when $\mathcal{P}_0$ is continuous [Shapiro, 2017, §3.2]. Under mild assumptions, Esfahani and Kuhn [2018] show that DRO problems over Wasserstein balls can be reformulated as finite convex programs—in some cases even as linear programs. Besides the constrained formulation in (33), there also exists a penalized (or regularized) form of DRO [Sinha et al., 2018], which makes the optimization problem more tractable. In the past years, a series of SA methods have been proposed for empirical DRO with convex losses [Namkoong and Duchi, 2016], and DRO with convex loss [Levy et al., 2020] and non-convex losses [Jin et al., 2021, Qi et al., 2021, Rafique et al., 2022].

The main focus of this paper is the GDRO problem in (2)/(3), instead of the traditional DRO in (33). Sagawa et al. [2020] have applied SMD [Nemirovski et al., 2009] to (3), but only obtain a sub-optimal sample complexity of $O(m^2(\log m)/\epsilon^2)$, because of the large variance in their gradients. In the sequel, Haghtalab et al. [2022] and Soma et al. [2022] have tried to improve the sample complexity by reusing samples and applying techniques from MAB respectively, but their analysis suffers dependency issues. Carmon and Hausler [2022, Proposition 2] successfully established an $O(m(\log m)/\epsilon^2)$ sample complexity by combining SMD and gradient clipping, but their result holds only in expectation. To deal with heterogeneous noise in different distributions, Agarwal and Zhang [2022] propose a variant of GDRO named as minimax regret optimization (MRO), which replaces the risk $R_i(\mathbf{w})$ with "excess risk" $R_i(\mathbf{w}) - \min_{\mathbf{w} \in \mathcal{W}} R_i(\mathbf{w})$. More generally, we can introduce calibration terms in DRO to prevent any single distribution to dominate the maximum [Słowik and Bottou, 2022].

Finally, we note that GDRO has a similar spirit with collaborative PAC learning [Blum et al., 2017, Nguyen and Zakynthinou, 2018, Rothblum and Yona, 2021] in the sense that both aim to find a single model that performs well on multiple distributions.

## B  Analysis

In this section, we present proofs of main theorems.

### B.1  Proof of Theorem 1

The proof is based on Lemma 3.1 and Proposition 3.2 of Nemirovski et al. [2009]. To apply them, we show that their preconditions are satisfied under our assumptions.

Although two instances of SMD are invoked to update $\mathbf{w}$ and $\mathbf{q}$ separately, they can be merged as 1 instance by concatenating $\mathbf{w}$ and $\mathbf{q}$ as a single variable $[\mathbf{w}; \mathbf{q}] \in \mathcal{W} \times \Delta_m$, and redefine the norm and the distance-generating function [Nemirovski et al., 2009, §3.1]. Let $\mathcal{E}$ be the space that $\mathcal{W}$ lies in. We equip the Cartesian product $\mathcal{E} \times \mathbb{R}^m$ with the following norm and dual norm:

$$\left\| [\mathbf{w}; \mathbf{q}] \right\| = \sqrt{\frac{1}{2D^2} \|\mathbf{w}\|_w^2 + \frac{1}{2\ln m} \|\mathbf{q}\|_1^2}, \text{ and } \left\| [\mathbf{u}; \mathbf{v}] \right\|_* = \sqrt{2D^2 \|\mathbf{u}\|_{w,*}^2 + 2\|\mathbf{v}\|_\infty^2 \ln m}. \quad (34)$$

We use the notation $\mathbf{x} = [\mathbf{w}; \mathbf{q}]$, and equip the set $\mathcal{W} \times \Delta_m$ with the distance-generating function

$$\nu(\mathbf{x}) = \nu([\mathbf{w}; \mathbf{q}]) = \frac{1}{2D^2} \nu_w(\mathbf{w}) + \frac{1}{2\ln m} \nu_q(\mathbf{q}). \quad (35)$$

It is easy to verify that $\nu(\mathbf{x})$ is 1-strongly convex w.r.t. the norm $\|\cdot\|$. Let $B(\cdot, \cdot)$ be the Bregman distance associated with $\nu(\cdot)$:

$$\begin{aligned} B(\mathbf{x}, \mathbf{x}') =& \nu(\mathbf{x}) - \left[ \nu(\mathbf{x}') + \langle \nabla\nu(\mathbf{x}'), \mathbf{x} - \mathbf{x}' \rangle \right] \\ =& \frac{1}{2D^2} \left( \nu_w(\mathbf{w}) - \left[ \nu_w(\mathbf{w}') + \langle \nabla\nu_w(\mathbf{w}'), \mathbf{w} - \mathbf{w}' \rangle \right] \right) \\ &+ \frac{1}{2\ln m} \left( \nu_q(\mathbf{q}) - \left[ \nu_q(\mathbf{q}') + \langle \nabla\nu_q(\mathbf{q}'), \mathbf{q} - \mathbf{q}' \rangle \right] \right) \\ =& \frac{1}{2D^2} B_w(\mathbf{w}, \mathbf{w}') + \frac{1}{2\ln m} B_q(\mathbf{q}, \mathbf{q}') \end{aligned} \quad (36)$$

where $\mathbf{x}' = [\mathbf{w}'; \mathbf{q}']$.

Then, we consider the following version of SMD for updating $\mathbf{x}_t$:

$$\mathbf{x}_{t+1} = \operatorname*{argmin}_{\mathbf{x} \in \mathcal{W} \times \Delta_m} \left\{ \eta \langle [\mathbf{g}_w(\mathbf{w}_t, \mathbf{q}_t); -\mathbf{g}_q(\mathbf{w}_t, \mathbf{q}_t)], \mathbf{x} - \mathbf{x}_t \rangle + B(\mathbf{x}, \mathbf{x}_t) \right\} \quad (37)$$

where $\eta > 0$ is the step size. In the beginning, we set $\mathbf{x}_1 = \operatorname{argmin}_{\mathbf{x} \in \mathcal{W} \times \Delta_m} \nu(\mathbf{x}) = [\mathbf{w}_1; \mathbf{q}_1]$. From the decomposition of the Bregman distance in (36), we observe that (37) is equivalent to (11) and (12) by setting

$$\eta_w = 2\eta D^2, \text{ and } \eta_q = 2\eta \ln m.$$

Next, we show that the stochastic gradients are well-bounded. Under our assumptions, we have

$$\|\mathbf{g}_w(\mathbf{w}_t, \mathbf{q}_t)\|_{w,*} = \left\| \sum_{i=1}^m q_{t,i} \nabla\ell(\mathbf{w}_t; \mathbf{z}_t^{(i)}) \right\|_{w,*} \leq \sum_{i=1}^m q_{t,i} \left\| \nabla\ell(\mathbf{w}_t; \mathbf{z}_t^{(i)}) \right\|_{w,*} \overset{(6)}{\leq} \sum_{i=1}^m q_{t,i} G = G,$$

$$\|\mathbf{g}_q(\mathbf{w}_t, \mathbf{q}_t)\|_\infty = \left\| [\ell(\mathbf{w}_t; \mathbf{z}_t^{(1)}), \dots, \ell(\mathbf{w}_t; \mathbf{z}_t^{(m)})]^\top \right\|_\infty \overset{(5)}{\leq} 1.$$

As a result, the concatenated gradients used in (37) is also bounded in term of the dual norm $\|\cdot\|_*$:

$$\begin{aligned} \left\| [\mathbf{g}_w(\mathbf{w}_t, \mathbf{q}_t); -\mathbf{g}_q(\mathbf{w}_t, \mathbf{q}_t)] \right\|_* =& \sqrt{2D^2 \|\mathbf{g}_w(\mathbf{w}_t, \mathbf{q}_t)\|_{w,*}^2 + 2\|\mathbf{g}_q(\mathbf{w}_t, \mathbf{q}_t)\|_\infty^2 \ln m} \\ \leq& \underbrace{\sqrt{2D^2 G^2 + 2\ln m}}_{:=M}. \end{aligned}$$

Now, we are ready to state our theoretical guarantees. By setting

$$\eta = \frac{2}{M\sqrt{5T}} = \sqrt{\frac{2}{5T(D^2 G^2 + \ln m)}},$$

Lemma 3.1 and (3.13) of Nemirovski et al. [2009] imply that

$$\mathrm{E}\big[\epsilon_\phi(\bar{\mathbf{w}}, \bar{\mathbf{q}})\big] \le 2M\sqrt{\frac{5}{T}} = 2\sqrt{\frac{10(D^2G^2 + \ln m)}{T}}$$

Furthermore, from Proposition 3.2 of Nemirovski et al. [2009], we have, for any $\Omega > 1$

$$\Pr\left[\epsilon_\phi(\bar{\mathbf{w}}, \bar{\mathbf{q}}) \ge (8 + 2\Omega)M\sqrt{\frac{5}{T}} = (8 + 2\Omega)\sqrt{\frac{10(D^2G^2 + \ln m)}{T}}\right] \le 2\exp(-\Omega).$$

We complete the proof by setting $\delta = 2\exp(-\Omega)$.

## B.2 Proof of Theorem 2

We first bound the regret of the 1st player. In the analysis, we address the non-obliviousness by the "ghost iterate" technique of Nemirovski et al. [2009].

**Theorem 5** *Under Assumptions 1, 2 and 4, by setting $\eta_w = \frac{2D}{G\sqrt{5T}}$, we have*

$$\mathrm{E}\left[\sum_{t=1}^T \phi(\mathbf{w}_t, \mathbf{q}_t) - \min_{\mathbf{w} \in \mathcal{W}} \sum_{t=1}^T \phi(\mathbf{w}, \mathbf{q}_t)\right] \le 2DG\sqrt{5T}$$

*and with probability at least $1 - \delta$,*

$$\sum_{t=1}^T \phi(\mathbf{w}_t, \mathbf{q}_t) - \min_{\mathbf{w} \in \mathcal{W}} \sum_{t=1}^T \phi(\mathbf{w}, \mathbf{q}_t) \le DG\sqrt{T}\left(2\sqrt{5} + 8\sqrt{\ln\frac{1}{\delta}}\right).$$

By extending Exp3-IX to stochastic rewards, we have the following bound for the 2nd player.

**Theorem 6** *Under Assumption 3, by setting $\eta_q = \sqrt{\frac{\ln m}{mT}}$ and the IX coefficient $\gamma = \frac{\eta_q}{2}$, we have*

$$\mathrm{E}\left[\max_{q \in \Delta_m} \sum_{t=1}^T \phi(\mathbf{w}_t, \mathbf{q}) - \sum_{t=1}^T \phi(\mathbf{w}_t, \mathbf{q}_t)\right] \le 3\sqrt{mT\ln m} + \sqrt{\frac{T}{2}} + 3\left(\sqrt{\frac{mT}{\ln m}} + \sqrt{\frac{T}{2}} + 1\right)$$

*and with probability at least $1 - \delta$,*

$$\max_{q \in \Delta_m} \sum_{t=1}^T \phi(\mathbf{w}_t, \mathbf{q}) - \sum_{t=1}^T \phi(\mathbf{w}_t, \mathbf{q}_t) \le 3\sqrt{mT\ln m} + \sqrt{\frac{T}{2}} + \left(\sqrt{\frac{mT}{\ln m}} + \sqrt{\frac{T}{2}} + 1\right)\ln\frac{3}{\delta}.$$

From Jensen's inequality and the outputs $\bar{\mathbf{w}} = \frac{1}{T}\sum_{t=1}^T \mathbf{w}_t$ and $\bar{\mathbf{q}} = \frac{1}{T}\sum_{t=1}^T \mathbf{q}_t$, we have

$$\epsilon_\phi(\bar{\mathbf{w}}, \bar{\mathbf{q}}) = \max_{\mathbf{q} \in \Delta_m} \phi(\bar{\mathbf{w}}, \mathbf{q}) - \min_{\mathbf{w} \in \mathcal{W}} \phi(\mathbf{w}, \bar{\mathbf{q}})$$

$$\le \frac{1}{T}\left(\max_{\mathbf{q} \in \Delta_m} \sum_{t=1}^T \phi(\mathbf{w}_t, \mathbf{q}) - \min_{\mathbf{w} \in \mathcal{W}} \sum_{t=1}^T \phi(\mathbf{w}, \mathbf{q}_t)\right) \tag{38}$$

$$= \frac{1}{T}\left(\max_{\mathbf{q} \in \Delta_m} \sum_{t=1}^T \phi(\mathbf{w}_t, \mathbf{q}) - \sum_{t=1}^T \phi(\mathbf{w}_t, \mathbf{q}_t)\right) + \frac{1}{T}\left(\sum_{t=1}^T \phi(\mathbf{w}_t, \mathbf{q}_t) - \min_{\mathbf{w} \in \mathcal{W}} \sum_{t=1}^T \phi(\mathbf{w}, \mathbf{q}_t)\right).$$

We obtain (19) by substituting the high probability bound in Theorems 5 and 6 into (38), and taking the union bound.

## B.3 Proof of Theorem 5

Our goal is to analyze SMD for non-oblivious OCO with stochastic gradients. In the literature, we did not find a convenient reference for it. A very close one is the Lemma 3.2 of Flaxman et al. [2005], which bounds the expected regret of SGD for non-oblivious OCO. But it is insufficient for our purpose, so we provide our proof by following the analysis of SMD for stochastic convex-concave

optimization [Nemirovski et al., 2009, §3]. Notice that we cannot use the theoretical guarantee of SMD for SCO [Nemirovski et al., 2009, §2.3], because the objective function is fixed in SCO.

From the standard analysis of mirror descent, e.g., Lemma 2.1 of Nemirovski et al. [2009], we have

$$\langle \tilde{\mathbf{g}}_w(\mathbf{w}_t, \mathbf{q}_t), \mathbf{w}_t - \mathbf{w} \rangle \leq \frac{B_w(\mathbf{w}, \mathbf{w}_t) - B_w(\mathbf{w}, \mathbf{w}_{t+1})}{\eta_w} + \frac{\eta_w}{2} \|\tilde{\mathbf{g}}_w(\mathbf{w}_t, \mathbf{q}_t)\|_{w,*}^2.$$

Summing the above inequality over $t = 1, \ldots, T$, we have

$$
\begin{aligned}
\sum_{t=1}^{T} \langle \tilde{\mathbf{g}}_w(\mathbf{w}_t, \mathbf{q}_t), \mathbf{w}_t - \mathbf{w} \rangle \leq & \frac{B_w(\mathbf{w}, \mathbf{w}_1)}{\eta_w} + \frac{\eta_w}{2} \sum_{t=1}^{T} \|\tilde{\mathbf{g}}_w(\mathbf{w}_t, \mathbf{q}_t)\|_{w,*}^2 \\
\overset{(6),(10)}{\leq} & \frac{B_w(\mathbf{w}, \mathbf{w}_1)}{\eta_w} + \frac{\eta_w T G^2}{2} \leq \frac{D^2}{\eta_w} + \frac{\eta_w T G^2}{2}
\end{aligned}
\tag{39}
$$

where the last step is due to [Nemirovski et al., 2009, (2.42)]

$$\max_{\mathbf{w} \in \mathcal{W}} B_w(\mathbf{w}, \mathbf{w}_1) \leq \max_{\mathbf{w} \in \mathcal{W}} \nu_w(\mathbf{w}) - \min_{\mathbf{w} \in \mathcal{W}} \nu_w(\mathbf{w}) \overset{(4)}{\leq} D^2. \tag{40}$$

By Jensen's inequality, we have

$$
\begin{aligned}
\sum_{t=1}^{T} [\phi(\mathbf{w}_t, \mathbf{q}_t) - \phi(\mathbf{w}, \mathbf{q}_t)] & \leq \sum_{t=1}^{T} \langle \nabla_{\mathbf{w}} \phi(\mathbf{w}_t, \mathbf{q}_t), \mathbf{w}_t - \mathbf{w} \rangle \\
& = \sum_{t=1}^{T} \langle \tilde{\mathbf{g}}_w(\mathbf{w}_t, \mathbf{q}_t), \mathbf{w}_t - \mathbf{w} \rangle + \sum_{t=1}^{T} \langle \nabla_{\mathbf{w}} \phi(\mathbf{w}_t, \mathbf{q}_t) - \tilde{\mathbf{g}}_w(\mathbf{w}_t, \mathbf{q}_t), \mathbf{w}_t - \mathbf{w} \rangle \\
& \overset{(39)}{\leq} \frac{D^2}{\eta_w} + \frac{\eta_w T G^2}{2} + \sum_{t=1}^{T} \langle \nabla_{\mathbf{w}} \phi(\mathbf{w}_t, \mathbf{q}_t) - \tilde{\mathbf{g}}_w(\mathbf{w}_t, \mathbf{q}_t), \mathbf{w}_t - \mathbf{w} \rangle.
\end{aligned}
$$

Maximizing each side over $\mathbf{w} \in \mathcal{W}$, we arrive at

$$
\begin{aligned}
\max_{\mathbf{w} \in \mathcal{W}} \sum_{t=1}^{T} [\phi(\mathbf{w}_t, \mathbf{q}_t) - \phi(\mathbf{w}, \mathbf{q}_t)] & = \sum_{t=1}^{T} \phi(\mathbf{w}_t, \mathbf{q}_t) - \min_{\mathbf{w} \in \mathcal{W}} \sum_{t=1}^{T} \phi(\mathbf{w}, \mathbf{q}_t) \\
& \leq \frac{D^2}{\eta_w} + \frac{\eta_w T G^2}{2} + \max_{\mathbf{w} \in \mathcal{W}} \left\{ \underbrace{\sum_{t=1}^{T} \langle \nabla_{\mathbf{w}} \phi(\mathbf{w}_t, \mathbf{q}_t) - \tilde{\mathbf{g}}_w(\mathbf{w}_t, \mathbf{q}_t), \mathbf{w}_t - \mathbf{w} \rangle}_{:= F(\mathbf{w})} \right\}.
\end{aligned}
\tag{41}
$$

Next, we bound the last term in (41), i.e., $\max_{\mathbf{w} \in \mathcal{W}} F(\mathbf{w})$. Because $\mathrm{E}_{t-1}[\tilde{\mathbf{g}}_w(\mathbf{w}_t, \mathbf{q}_t)] = \nabla_{\mathbf{w}} \phi(\mathbf{w}_t, \mathbf{q}_t)$, $F(\mathbf{w})$ is the sum of a martingale difference sequence for any *fixed* $\mathbf{w}$. However, it is not true for $\tilde{\mathbf{w}} = \operatorname{argmax}_{\mathbf{w} \in \mathcal{W}} F(\mathbf{w})$, because $\tilde{\mathbf{w}}$ depends on the randomness of the algorithm. Thus, we cannot directly apply techniques for martingales to bounding $\max_{\mathbf{w} \in \mathcal{W}} F(\mathbf{w})$. This is the place where the analysis differs from that of SCO.

To handle the above challenge, we introduce a virtual sequence of variables to decouple the dependency [Nemirovski et al., 2009, proof of Lemma 3.1]. Imagine there is an online algorithm which performs SMD by using $\nabla_{\mathbf{w}} \phi(\mathbf{w}_t, \mathbf{q}_t) - \tilde{\mathbf{g}}_w(\mathbf{w}_t, \mathbf{q}_t)$ as the gradient:

$$\mathbf{v}_{t+1} = \operatorname*{argmin}_{\mathbf{w} \in \mathcal{W}} \left\{ \eta_w \langle \nabla_{\mathbf{w}} \phi(\mathbf{w}_t, \mathbf{q}_t) - \tilde{\mathbf{g}}_w(\mathbf{w}_t, \mathbf{q}_t), \mathbf{w} - \mathbf{v}_t \rangle + B_w(\mathbf{w}, \mathbf{v}_t) \right\} \tag{42}$$

where $\mathbf{v}_1 = \mathbf{w}_1$. By repeating the derivation of (39), we can show that

$$
\begin{aligned}
& \sum_{t=1}^{T} \langle \nabla_{\mathbf{w}} \phi(\mathbf{w}_t, \mathbf{q}_t) - \tilde{\mathbf{g}}_w(\mathbf{w}_t, \mathbf{q}_t), \mathbf{v}_t - \mathbf{w} \rangle \\
& \leq \frac{B_w(\mathbf{w}, \mathbf{w}_1)}{\eta_w} + \frac{\eta_w}{2} \sum_{t=1}^{T} \|\nabla_{\mathbf{w}} \phi(\mathbf{w}_t, \mathbf{q}_t) - \tilde{\mathbf{g}}_w(\mathbf{w}_t, \mathbf{q}_t)\|_{w,*}^2 \leq \frac{D^2}{\eta_w} + 2\eta_w T G^2
\end{aligned}
\tag{43}
$$

where in the last inequality, we make use of (40) and

$$\|\nabla_{\mathbf{w}}\phi(\mathbf{w}_t, \mathbf{q}_t) - \tilde{\mathbf{g}}_w(\mathbf{w}_t, \mathbf{q}_t)\|_{w,*} \leq \|\phi(\mathbf{w}_t, \mathbf{q}_t)\|_{w,*} + \|\tilde{\mathbf{g}}_w(\mathbf{w}_t, \mathbf{q}_t)\|_{w,*}$$

$$\leq \mathrm{E}_{t-1}[\|\tilde{\mathbf{g}}_w(\mathbf{w}_t, \mathbf{q}_t)\|_{w,*}] + \|\tilde{\mathbf{g}}_w(\mathbf{w}_t, \mathbf{q}_t)\|_{w,*} \overset{(6),(10)}{\leq} 2G. \tag{44}$$

Then, we have

$$\max_{\mathbf{w} \in \mathcal{W}} \left\{ \sum_{t=1}^{T} \langle \nabla_{\mathbf{w}}\phi(\mathbf{w}_t, \mathbf{q}_t) - \tilde{\mathbf{g}}_w(\mathbf{w}_t, \mathbf{q}_t), \mathbf{w}_t - \mathbf{w} \rangle \right\}$$

$$= \max_{\mathbf{w} \in \mathcal{W}} \left\{ \sum_{t=1}^{T} \langle \nabla_{\mathbf{w}}\phi(\mathbf{w}_t, \mathbf{q}_t) - \tilde{\mathbf{g}}_w(\mathbf{w}_t, \mathbf{q}_t), \mathbf{v}_t - \mathbf{w} \rangle \right\} + \sum_{t=1}^{T} \langle \nabla_{\mathbf{w}}\phi(\mathbf{w}_t, \mathbf{q}_t) - \tilde{\mathbf{g}}_w(\mathbf{w}_t, \mathbf{q}_t), \mathbf{w}_t - \mathbf{v}_t \rangle$$

$$\overset{(43)}{\leq} \frac{D^2}{\eta_w} + 2\eta_w T G^2 + \sum_{t=1}^{T} \underbrace{\langle \nabla_{\mathbf{w}}\phi(\mathbf{w}_t, \mathbf{q}_t) - \tilde{\mathbf{g}}_w(\mathbf{w}_t, \mathbf{q}_t), \mathbf{w}_t - \mathbf{v}_t \rangle}_{V_t}. \tag{45}$$

From the updating rule of $\mathbf{v}_t$ in (42), we know that $\mathbf{v}_t$ is independent from $\nabla_{\mathbf{w}}\phi(\mathbf{w}_t, \mathbf{q}_t) - \tilde{\mathbf{g}}_w(\mathbf{w}_t, \mathbf{q}_t)$, and thus $V_1, \ldots, V_T$ is a martingale difference sequence.

Substituting (45) into (41), we have

$$\sum_{t=1}^{T} \phi(\mathbf{w}_t, \mathbf{q}_t) - \min_{\mathbf{w} \in \mathcal{W}} \sum_{t=1}^{T} \phi(\mathbf{w}, \mathbf{q}_t) \leq \frac{2D^2}{\eta_w} + \frac{5\eta_w T G^2}{2} + \sum_{t=1}^{T} V_t. \tag{46}$$

Taking expectation over both sides, we have

$$\mathrm{E}\left[ \sum_{t=1}^{T} \phi(\mathbf{w}_t, \mathbf{q}_t) - \min_{\mathbf{w} \in \mathcal{W}} \sum_{t=1}^{T} \phi(\mathbf{w}, \mathbf{q}_t) \right] \leq \frac{2D^2}{\eta_w} + \frac{5\eta_w T G^2}{2} = 2DG\sqrt{5T}$$

where we set $\eta_w = \frac{2D}{G\sqrt{5T}}$.

To establish a high probability bound, we make use of the Hoeffding-Azuma inequality for martingales stated below [Cesa-Bianchi and Lugosi, 2006].

**Lemma 1** *Let* $V_1, V_2, \ldots$ *be a martingale difference sequence with respect to some sequence* $X_1, X_2, \ldots$ *such that* $V_i \in [A_i, A_i + c_i]$ *for some random variable* $A_i$, *measurable with respect to* $X_1, \ldots, X_{i-1}$ *and a positive constant* $c_i$. *If* $S_n = \sum_{i=1}^{n} V_i$, *then for any* $t > 0$,

$$\mathrm{Pr}[S_n > t] \leq \exp\left( -\frac{2t^2}{\sum_{i=1}^{n} c_i^2} \right).$$

To apply the above lemma, we need to show that $V_t$ is bounded. Indeed, we have

$$|\langle \nabla_{\mathbf{w}}\phi(\mathbf{w}_t, \mathbf{q}_t) - \tilde{\mathbf{g}}_w(\mathbf{w}_t, \mathbf{q}_t), \mathbf{w}_t - \mathbf{v}_t \rangle| \leq \|\nabla_{\mathbf{w}}\phi(\mathbf{w}_t, \mathbf{q}_t) - \tilde{\mathbf{g}}_w(\mathbf{w}_t, \mathbf{q}_t)\|_{w,*} \|\mathbf{w}_t - \mathbf{v}_t\|_w$$

$$\overset{(44)}{\leq} 2G\|\mathbf{w}_t - \mathbf{v}_t\|_w \leq 2G\left( \|\mathbf{w}_t - \mathbf{w}_1\|_w + \|\mathbf{v}_t - \mathbf{w}_1\|_w \right)$$

$$\leq 2G\left( \sqrt{2B_w(\mathbf{w}_t, \mathbf{w}_1)} + \sqrt{2B_w(\mathbf{v}_t, \mathbf{w}_1)} \right) \overset{(40)}{\leq} 4\sqrt{2}DG.$$

From Lemma 1, with probability at least $1 - \delta$, we have

$$\sum_{t=1}^{T} V_t \leq 8DG\sqrt{T \ln \frac{1}{\delta}}. \tag{47}$$

We complete the proof by substituting (47) into (46).

## B.4 Proof of Theorem 6

Because we can only observe $\ell(\mathbf{w}_t, \mathbf{z}_t^{(i_t)})$ instead of $R_{i_t}(\mathbf{w}_t)$, the theoretical guarantee of Exp3-IX [Neu, 2015] cannot be directly applied to Algorithm 2. To address this challenge, we generalize the regret analysis of Exp3-IX to stochastic rewards.

By the definition of $\phi(\mathbf{w}, \mathbf{q})$ in (3) and the property of linear optimization over the simplex, we have

$$
\max_{\mathbf{q}\in\Delta_m} \sum_{t=1}^{T} \phi(\mathbf{w}_t, \mathbf{q}) - \sum_{t=1}^{T} \phi(\mathbf{w}_t, \mathbf{q}_t) = \max_{\mathbf{q}\in\Delta_m} \sum_{i=1}^{m} q_i \left( \sum_{t=1}^{T} R_i(\mathbf{w}_t) \right) - \sum_{t=1}^{T} \sum_{i=1}^{m} q_{t,i} R_i(\mathbf{w}_t)
$$

$$
= \sum_{t=1}^{T} R_{j^*}(\mathbf{w}_t) - \sum_{t=1}^{T} \sum_{i=1}^{m} q_{t,i} R_i(\mathbf{w}_t) = \sum_{t=1}^{T} \mathbb{E}_{\mathbf{z}\sim\mathcal{P}_{j^*}}[\ell(\mathbf{w}_t; \mathbf{z})] - \sum_{t=1}^{T} \sum_{i=1}^{m} q_{t,i} \mathbb{E}_{\mathbf{z}\sim\mathcal{P}_i}[\ell(\mathbf{w}_t; \mathbf{z})] \quad (48)
$$

$$
= \sum_{t=1}^{T} \sum_{i=1}^{m} q_{t,i} s_{t,i} - \sum_{t=1}^{T} s_{t,j^*} = \sum_{t=1}^{T} \langle \mathbf{q}_t, \mathbf{s}_t \rangle - \sum_{t=1}^{T} s_{t,j^*}
$$

where $j^* \in \operatorname{argmax}_{j\in[m]} \sum_{t=1}^{T} R_j(\mathbf{w}_t)$ and the vector $\mathbf{s}_t \in \mathbb{R}^m$ is defined as

$$
s_{t,i} \triangleq 1 - \mathbb{E}_{\mathbf{z}\sim\mathcal{P}_i}[\ell(\mathbf{w}_t; \mathbf{z})] \overset{(5)}{\in} [0,1], \ \forall i \in [m].
$$

To facilitate the analysis, we introduce a vector $\hat{\mathbf{s}}_t \in \mathbb{R}^m$ with

$$
\hat{s}_{t,i} \triangleq 1 - \ell(\mathbf{w}_t; \mathbf{z}_t^{(i)}) \overset{(5)}{\in} [0,1], \ \forall i \in [m] \quad (49)
$$

where $\mathbf{z}_t^{(i)}$ denotes a random sample drawn from the $i$-th distribution. Note that $\hat{\mathbf{s}}_t$ is only used for *analysis*, with the purpose of handling the stochastic rewards. In the algorithm, only $\hat{s}_{t,i_t} = 1 - \ell(\mathbf{w}_t; \mathbf{z}_t^{(i_t)})$ is observed in the $t$-th iteration.

Following the proof of Theorem 1 of Neu [2015], we have

$$
\sum_{t=1}^{T} \langle \mathbf{q}_t, \tilde{\mathbf{s}}_t \rangle - \sum_{t=1}^{T} \tilde{s}_{t,j^*} \leq \frac{\ln m}{\eta_q} + \frac{\eta_q}{2} \sum_{t=1}^{T} \sum_{i=1}^{m} \tilde{s}_{t,i} \quad (50)
$$

which makes use of the property of online mirror descent with local norms [Bubeck and Cesa-Bianchi, 2012, Theorem 5.5]. From (5) of Neu [2015], we have

$$
\langle \mathbf{q}_t, \tilde{\mathbf{s}}_t \rangle = \sum_{i=1}^{m} q_{t,i} \tilde{s}_{t,i} = \hat{s}_{t,i_t} - \gamma \sum_{i=1}^{m} \tilde{s}_{t,i}. \quad (51)
$$

Combining (50) and (51), we have

$$
\sum_{t=1}^{T} \hat{s}_{t,i_t} \leq \sum_{t=1}^{T} \tilde{s}_{t,j^*} + \left( \frac{\eta_q}{2} + \gamma \right) \sum_{t=1}^{T} \sum_{i=1}^{m} \tilde{s}_{t,i} + \frac{\ln m}{\eta_q}. \quad (52)
$$

From (48), we have

$$
\max_{\mathbf{q}\in\Delta_m} \sum_{t=1}^{T} \phi(\mathbf{w}_t, \mathbf{q}) - \sum_{t=1}^{T} \phi(\mathbf{w}_t, \mathbf{q}_t)
$$

$$
= \sum_{t=1}^{T} \hat{s}_{t,i_t} - \sum_{t=1}^{T} s_{t,j^*} + \sum_{t=1}^{T} \langle \mathbf{q}_t, \mathbf{s}_t \rangle - \sum_{t=1}^{T} \hat{s}_{t,i_t} \quad (53)
$$

$$
\overset{(52)}{\leq} \underbrace{\sum_{t=1}^{T} \left( \tilde{s}_{t,j^*} - s_{t,j^*} \right)}_{:=A} + \underbrace{\left( \frac{\eta_q}{2} + \gamma \right) \sum_{t=1}^{T} \sum_{i=1}^{m} \tilde{s}_{t,i}}_{:=B} + \underbrace{\sum_{t=1}^{T} \left( \langle \mathbf{q}_t, \mathbf{s}_t \rangle - \hat{s}_{t,i_t} \right)}_{:=C} + \frac{\ln m}{\eta_q}.
$$

We proceed to bound the above three terms $A$, $B$ and $C$ respectively.

To bound term $A$, we need the following concentration result concerning the IX loss estimates [Neu, 2015, Lemma 1], which we further generalize to the setting with stochastic rewards.

**Lemma 2** *Let $\xi_{t,i} \in [0,1]$ for all $t \in [T]$ and $i \in [m]$, and $\tilde{\xi}_{t,i}$ be its IX-estimator defined as $\tilde{\xi}_{t,i} = \frac{\hat{\xi}_{t,i}}{p_{t,i}+\gamma_t}\mathbb{I}[i_t = i]$, where $\hat{\xi}_{t,i}$ is such that $\mathrm{E}[\hat{\xi}_{t,i}] = \xi_{t,i}$ with $\hat{\xi}_{t,i} \in [0,1]$ and the index $i_t$ is sampled from $[m]$ according to the distribution $\mathbf{p}_t \in \Delta_m$. Let $\{\gamma_t\}_{t=1}^T$ be a fixed non-increasing sequence with $\gamma_t \geq 0$ and let $\alpha_{t,i}$ be non-negative $\mathcal{F}_{t-1}$-measurable random variables satisfying $\alpha_{t,i} \leq 2\gamma_t$ for all $t \in [T]$ and $i \in [m]$. Then, for any $\delta > 0$, with probability at least $1 - \delta$,*

$$\sum_{t=1}^{T}\sum_{i=1}^{m}\alpha_{t,i}(\tilde{\xi}_{t,i} - \xi_{t,i}) \leq \ln\frac{1}{\delta}. \tag{54}$$

*Furthermore, when $\gamma_t = \gamma \geq 0$ for all $t \in [T]$, the following holds with probability at least $1 - \delta$,*

$$\sum_{t=1}^{T}(\tilde{\xi}_{t,i} - \xi_{t,i}) \leq \frac{1}{2\gamma}\ln\frac{m}{\delta} \tag{55}$$

*simultaneously for all $i \in [m]$.*

Notice that our construction of $\tilde{\mathbf{s}}_t$ in (17) satisfies that $\tilde{s}_{t,i} = \frac{\hat{s}_{t,i}}{q_{t,i}+\gamma}\mathbb{I}[i_t = i]$ and $i_t$ is drawn from $[m]$ according to $\mathbf{q}_t \in \Delta_m$ as well as $\mathrm{E}[\hat{s}_{t,i}] = s_{t,i}$, which meets the conditions required by Lemma 2. As a result, according to (55), we have

$$\sum_{t=1}^{T}(\tilde{s}_{t,j} - s_{t,j}) \leq \frac{1}{2\gamma}\ln\frac{m}{\delta}$$

for all $j \in [m]$ (including $j^*$) with probability at least $1 - \delta$.

To bound term $B$, we can directly use Lemma 1 of Neu [2015], because our setting $\frac{\eta_q}{2} = \gamma$ satisfies its requirement. Thus, with probability at least $1 - \delta$, we have

$$\left(\frac{\eta_q}{2} + \gamma\right)\sum_{t=1}^{T}\sum_{i=1}^{m}\tilde{s}_{t,j} \leq \left(\frac{\eta_q}{2} + \gamma\right)\sum_{t=1}^{T}\sum_{i=1}^{m}\hat{s}_{t,j} + \ln\frac{1}{\delta} \overset{(49)}{\leq} \left(\frac{\eta_q}{2} + \gamma\right)mT + \ln\frac{1}{\delta}. \tag{56}$$

We now consider term $C$ in (53). Let $V_t = \langle \mathbf{q}_t, \mathbf{s}_t \rangle - \hat{s}_{t,i_t}$. Then, it is easy to verify that $\mathrm{E}_{t-1}[V_t] = 0$. So, the process $\{V_t\}_{t=1}^T$ forms a martingale difference sequence and it also satisfies $|V_t| \leq 1$ for all $t$. Hence, we can apply Lemma 1 and have

$$\sum_{t=1}^{T}\left(\langle \mathbf{q}_t, \mathbf{s}_t \rangle - \hat{s}_{t,i_t}\right) \leq \sqrt{2T\ln\frac{1}{\delta}} \leq \sqrt{\frac{T}{2}\left(1 + \ln\frac{1}{\delta}\right)}.$$

with probability at least $1 - \delta$.

Combining the three upper bounds for the terms $A$, $B$ and $C$, and further taking the union bound, we have, with probability at least $1 - \delta$

$$\max_{q \in \Delta_m}\sum_{t=1}^{T}\phi(\mathbf{w}_t, \mathbf{q}) - \sum_{t=1}^{T}\phi(\mathbf{w}_t, \mathbf{q}_t)$$

$$\leq \frac{1}{2\gamma}\ln\frac{3m}{\delta} + \left(\frac{\eta_q}{2} + \gamma\right)mT + \ln\frac{3}{\delta} + \sqrt{\frac{T}{2}\left(1 + \ln\frac{3}{\delta}\right)} + \frac{\ln m}{\eta_q}$$

$$= 2\sqrt{mT\ln m} + \sqrt{\frac{mT}{\ln m}}\cdot\ln\frac{3m}{\delta} + \sqrt{\frac{T}{2}} + \left(\sqrt{\frac{T}{2}} + 1\right)\ln\frac{3}{\delta}$$

$$= 3\sqrt{mT\ln m} + \sqrt{\frac{T}{2}} + \left(\sqrt{\frac{mT}{\ln m}} + \sqrt{\frac{T}{2}} + 1\right)\ln\frac{3}{\delta},$$

where the third line holds because of our parameter settings $\gamma = \frac{\eta_q}{2}$ and $\eta_q = \sqrt{\frac{\ln m}{mT}}$.

Based on this high probability guarantee, we can then obtain the expected regret upper bound using the formula that

$$\mathrm{E}[X] \leq \int_0^1 \frac{1}{\delta} \Pr\left[X > \ln\frac{1}{\delta}\right] \mathrm{d}\delta$$

holds for any real-valued random variable $X$ [Bubeck and Cesa-Bianchi, 2012, § 3.2]. In particular, taking

$$X = \left(\sqrt{\frac{mT}{\ln m}} + \sqrt{\frac{T}{2}} + 1\right)^{-1} \cdot \left(\max_{q \in \Delta_m} \sum_{t=1}^T \phi(\mathbf{w}_t, \mathbf{q}) - \sum_{t=1}^T \phi(\mathbf{w}_t, \mathbf{q}_t) - 3\sqrt{mT\ln m} - \sqrt{\frac{T}{2}}\right)$$

yields $\mathrm{E}[X] \leq 3$, which implies that

$$\mathrm{E}\left[\max_{q \in \Delta_m} \sum_{t=1}^T \phi(\mathbf{w}_t, \mathbf{q}) - \sum_{t=1}^T \phi(\mathbf{w}_t, \mathbf{q}_t)\right] \leq 3\sqrt{mT\ln m} + \sqrt{\frac{T}{2}} + 3\left(\sqrt{\frac{mT}{\ln m}} + \sqrt{\frac{T}{2}} + 1\right).$$

## B.5  Proof of Theorem 3

For the stochastic gradients in (22), their norm can be upper bounded in the same way as (9). That is,

$$\|\mathbf{g}_w(\mathbf{w}_t, \mathbf{q}_t)\|_{w,*} = \left\|\sum_{i \in C_t} q_{t,i} \nabla\ell(\mathbf{w}_t; \mathbf{z}_t^{(i)})\right\|_{w,*} \leq \sum_{i \in C_t} q_{t,i} \left\|\nabla\ell(\mathbf{w}_t; \mathbf{z}_t^{(i)})\right\|_{w,*} \overset{(6)}{\leq} \sum_{i \in C_t} q_{t,i} G = G,$$

$$\|\mathbf{g}_q(\mathbf{w}_t, \mathbf{q}_t)\|_{\infty} = \max_{i \in C_t} |\ell(\mathbf{w}_t; \mathbf{z}_t^{(i)})| \overset{(5)}{\leq} 1.$$

So, with exactly the same analysis as Theorem 1, we have

$$\mathrm{E}\big[\epsilon_\varphi(\bar{\mathbf{w}}, \bar{\mathbf{q}})\big] \leq 2\sqrt{\frac{10(D^2G^2 + \ln m)}{n_1}}$$

and with probability at least $1 - \delta$,

$$\epsilon_\varphi(\bar{\mathbf{w}}, \bar{\mathbf{q}}) \leq \left(8 + 2\ln\frac{2}{\delta}\right)\sqrt{\frac{10(D^2G^2 + \ln m)}{n_1}}. \tag{57}$$

Next, we discuss how to bound the risk of $\bar{\mathbf{w}}$ on every distribution $\mathcal{P}_i$, i.e., $R_i(\bar{\mathbf{w}})$. Following the derivation in (8), we know

$$\max_{i \in [m]} p_i R_i(\bar{\mathbf{w}}) - \min_{\mathbf{w} \in \mathcal{W}} \max_{i \in [m]} p_i R_i(\mathbf{w}) \leq \epsilon_\varphi(\bar{\mathbf{w}}, \bar{\mathbf{q}}).$$

Thus, for every distribution $\mathcal{P}_i$, $R_i(\bar{\mathbf{w}})$ can be bounded in the following way:

$$R_i(\bar{\mathbf{w}}) \leq \frac{1}{p_i} \min_{\mathbf{w} \in \mathcal{W}} \max_{i \in [m]} p_i R_i(\mathbf{w}) + \frac{1}{p_i} \epsilon_\varphi(\bar{\mathbf{w}}, \bar{\mathbf{q}}).$$

Taking the high probability bound in (57) as an example, we have with probability at $1 - \delta$

$$\begin{aligned}
R_i(\bar{\mathbf{w}}) &\leq \frac{1}{p_i} \min_{\mathbf{w} \in \mathcal{W}} \max_{i \in [m]} p_i R_i(\mathbf{w}) + \frac{1}{p_i}\left(8 + 2\ln\frac{2}{\delta}\right)\sqrt{\frac{10(D^2G^2 + \ln m)}{n_1}} \\
&= \frac{n_1}{n_i} \min_{\mathbf{w} \in \mathcal{W}} \max_{i \in [m]} p_i R_i(\mathbf{w}) + \left(8 + 2\ln\frac{2}{\delta}\right)\frac{\sqrt{10(D^2G^2 + \ln m)n_1}}{n_i}.
\end{aligned} \tag{58}$$

## B.6  Proof of Theorem 4

We first provide some simple facts that will be used later. From Assumption 3, we immediately know that each risk function $R_i(\cdot)$ also belongs to $[0, 1]$. As a result, the difference between each risk function and its estimator is well-bounded, i.e., for all $i \in [m]$,

$$-1 \leq R_i(\mathbf{w}) - \ell(\mathbf{w}; \mathbf{z}) \leq 1, \ \forall \mathbf{w} \in \mathcal{W}, \ \mathbf{z} \sim \mathcal{P}_i. \tag{59}$$

From Assumption 4, we can prove that each risk function $R_i(\cdot)$ is $G$-Lipschitz continuous. To see this, we have

$$\|\nabla R_i(\mathbf{w})\|_{w,*} = \|\mathrm{E}_{\mathbf{z}\sim\mathcal{P}_i}\nabla\ell(\mathbf{w};\mathbf{z})\|_{w,*} \leq \mathrm{E}_{\mathbf{z}\sim\mathcal{P}_i}\|\nabla\ell(\mathbf{w};\mathbf{z})\|_{w,*} \overset{(6)}{\leq} G, \ \forall\mathbf{w}\in\mathcal{W}, i\in[m]. \quad (60)$$

As a result, we have

$$|R_i(\mathbf{w}) - R_i(w')| \leq G\|\mathbf{w} - \mathbf{w}'\|_w, \ \forall\mathbf{w}, \mathbf{w}'\in\mathcal{W}, i\in[m]. \quad (61)$$

Furthermore, the difference between the gradient of $R_i(\cdot)$ and its estimator is also well-bounded, i.e., for all $i\in[m]$,

$$\|\nabla R_i(\mathbf{w}) - \nabla\ell(\mathbf{w};\mathbf{z})\|_{w,*} \leq \|\nabla R_i(\mathbf{w})\|_{w,*} + \|\nabla\ell(\mathbf{w};\mathbf{z})\|_{w,*} \overset{(6),(60)}{\leq} 2G, \ \forall\mathbf{w}\in\mathcal{W}, \mathbf{z}\sim\mathcal{P}_i. \quad (62)$$

Recall the definition of the norm $\|\cdot\|$ and dual norm $\|\cdot\|_*$ for the space $\mathcal{E}\times\mathbb{R}^m$ in (34), and the distance-generating function $\nu(\cdot)$ in (35). Following the arguments in Section B.1, the two updating rules in (25) and (26) can be merged as

$$[\mathbf{w}_{t+1};\mathbf{q}_{t+1}] = \underset{\mathbf{x}\in\mathcal{W}\times\Delta_m}{\operatorname{argmin}} \left\{\eta\langle[\mathbf{g}_w(\mathbf{w}'_t,\mathbf{q}'_t); -\mathbf{g}_q(\mathbf{w}'_t,\mathbf{q}'_t)], \mathbf{x} - [\mathbf{w}'_t;\mathbf{q}'_t]\rangle + B(\mathbf{x},[\mathbf{w}'_t;\mathbf{q}'_t])\right\} \quad (63)$$

where $\eta_w = 2\eta D^2$ and $\eta_q = 2\eta\ln m$. Similarly, (28) and (29) are equivalent to

$$[\mathbf{w}'_{t+1};\mathbf{q}'_{t+1}] = \underset{\mathbf{x}\in\mathcal{W}\times\Delta_m}{\operatorname{argmin}} \left\{\eta\langle[\mathbf{g}_w(\mathbf{w}_{t+1},\mathbf{q}_{t+1}); -\mathbf{g}_q(\mathbf{w}_{t+1},\mathbf{q}_{t+1})], \mathbf{x} - [\mathbf{w}'_t;\mathbf{q}'_t]\rangle + B(\mathbf{x},[\mathbf{w}'_t;\mathbf{q}'_t])\right\}. \quad (64)$$

Let $F([\mathbf{w};\mathbf{q}])$ be the monotone operator associated with the weighted GDRO problem in (23), i.e.,

$$\begin{aligned} F([\mathbf{w};\mathbf{q})] &= [\nabla_{\mathbf{w}}\varphi(\mathbf{w},\mathbf{q}); -\nabla_{\mathbf{q}}\varphi(\mathbf{w},\mathbf{q})] \\ &= \left[\sum_{i=1}^m q_i p_i \nabla R_i(\mathbf{w}); -[p_1 R_1(\mathbf{w}), \ldots, p_m R_m(\mathbf{w})]^\top\right]. \end{aligned} \quad (65)$$

From our constructions of stochastic gradients in (24) and (27), we clearly have

$$\begin{aligned} \mathrm{E}_{t-1}\left\{[\mathbf{g}_w(\mathbf{w}'_t,\mathbf{q}'_t); -\mathbf{g}_q(\mathbf{w}'_t,\mathbf{q}'_t)]\right\} &= F([\mathbf{w}'_t;\mathbf{q}'_t]), \\ \mathrm{E}_{t-1}\left\{[\mathbf{g}_w(\mathbf{w}_{t+1},\mathbf{q}_{t+1}); -\mathbf{g}_q(\mathbf{w}_{t+1},\mathbf{q}_{t+1})]\right\} &= F([\mathbf{w}_{t+1};\mathbf{q}_{t+1}]). \end{aligned}$$

Thus, Algorithm 4 is indeed an instance of SMPA [Juditsky et al., 2011, Algorithm 1], and we can use their Theorem 1 and Corollary 1 to bound the optimization error.

Before applying their results, we show that all the preconditions are satisfied. The parameter $\Omega$ defined in (16) of Juditsky et al. [2011] can be upper bounded by

$$\begin{aligned} \Omega &= \sqrt{2\max_{\mathbf{x}\in\mathcal{W}\times\Delta_m} B(\mathbf{x},[\mathbf{w}'_1;\mathbf{q}'_1])} \overset{(36)}{=} \sqrt{\frac{1}{D^2}\max_{\mathbf{w}\in\mathcal{W}} B_w(\mathbf{w}_1,\mathbf{w}'_1) + \max_{\mathbf{q}\in\Delta_m}\frac{1}{\ln m}B_q(\mathbf{q},\mathbf{q}'_1)} \\ &\overset{(40)}{\leq} \sqrt{\frac{1}{D^2}\left(\max_{\mathbf{w}\in\mathcal{W}}\nu_w(\mathbf{w}) - \min_{\mathbf{w}\in\mathcal{W}}\nu_w(\mathbf{w})\right) + \frac{1}{\ln m}\left(\max_{\mathbf{q}\in\Delta_m}\nu_q(\mathbf{q}) - \min_{\mathbf{q}\in\Delta_m}\nu_q(\mathbf{q})\right)} \overset{(4)}{=} \sqrt{2}. \end{aligned} \quad (66)$$

Next, we need to demonstrate that $F([\mathbf{w};\mathbf{q}])$ is continuous.

**Lemma 3** *For the monotone operator $F([\mathbf{w};\mathbf{q}])$, we have*

$$\|F([\mathbf{w};\mathbf{q}]) - F([\mathbf{w}';\mathbf{q}'])\|_* \leq \widetilde{L}\|[\mathbf{w} - \mathbf{w}';\mathbf{q} - \mathbf{q}']\|$$

*where $\widetilde{L}$ is defined in (31).*

We proceed to show the variance of the stochastic gradients satisfies the light tail condition. To this end, we introduce the stochastic oracle used in Algorithm 4:

$$\mathbf{g}([\mathbf{w};\mathbf{q}]) = [\mathbf{g}_w(\mathbf{w},\mathbf{q}); -\mathbf{g}_q(\mathbf{w},\mathbf{q})] \quad (67)$$

where

$$\mathbf{g}_w(\mathbf{w}, \mathbf{q}) = \sum_{i=1}^{m} q_i p_i \left( \frac{n_m}{n_i} \sum_{j=1}^{n_i/n_m} \nabla \ell(\mathbf{w}; \mathbf{z}^{(i,j)}) \right),$$

$$\mathbf{g}_q(\mathbf{w}, \mathbf{q}) = \left[ p_1 \frac{n_m}{n_1} \sum_{j=1}^{n_1/n_m} \ell(\mathbf{w}; \mathbf{z}^{(1,j)}), \dots, p_m \ell(\mathbf{w}; \mathbf{z}^{(m)}) \right]^{\top}$$

and $\mathbf{z}^{(i,j)}$ is the $j$-th sample drawn from distribution $\mathcal{P}_i$. The following lemma shows that the variance is indeed sub-Gaussian.

**Lemma 4** *For the stochastic oracle $\mathbf{g}([\mathbf{w}; \mathbf{q}])$, we have*

$$\mathrm{E}\left[ \exp\left( \frac{\|F([\mathbf{w}; \mathbf{q}]) - \mathbf{g}([\mathbf{w}; \mathbf{q}])\|_*^2}{\sigma^2} \right) \right] \le 2$$

*where $\sigma^2$ is defined in (31).*

Based on (66), Lemma 3, and Lemma 4, we can apply the theoretical guarantee of SMPA. Recall that the total number of iterations is $n_m/2$ in Algorithm 4. From Corollary 1 of Juditsky et al. [2011], by setting

$$\eta = \min\left( \frac{1}{\sqrt{3}\widetilde{L}}, \frac{2}{\sqrt{7\sigma^2 n_m}} \right)$$

we have

$$\Pr\left[ \epsilon_\varphi(\bar{\mathbf{w}}, \bar{\mathbf{q}}) \ge \frac{7\widetilde{L}}{n_m} + 14\sqrt{\frac{2\sigma^2}{3n_m}} + 7\Lambda\sqrt{\frac{\sigma^2}{n_m}} \right] \le \exp\left( -\frac{\Lambda^2}{3} \right) + \exp\left( -\frac{\Lambda n_m}{2} \right)$$

for all $\Lambda > 0$. Choosing $\Lambda$ such that $\exp(-\Lambda^2/3) \le \delta/2$ and $\exp(-\Lambda n_m/2) \le \delta/2$, we have with probability at least $1 - \delta$

$$\epsilon_\varphi(\bar{\mathbf{w}}, \bar{\mathbf{q}}) \le \frac{7\widetilde{L}}{n_m} + 14\sqrt{\frac{2\sigma^2}{3n_m}} + 7\left( \sqrt{3\log\frac{2}{\delta}} + \frac{2}{n_m}\log\frac{2}{\delta} \right)\sqrt{\frac{\sigma^2}{n_m}}.$$

Following the derivation of (58), we have

$$R_i(\bar{\mathbf{w}}) - \frac{1}{p_i} \min_{\mathbf{w} \in \mathcal{W}} \max_{i \in [m]} p_i R_i(\mathbf{w})$$
$$\le \frac{1}{p_i} \left( \frac{7\widetilde{L}}{n_m} + \sqrt{\frac{\sigma^2}{n_m}} \left( 14\sqrt{\frac{2}{3}} + 7\sqrt{3\log\frac{2}{\delta}} + \frac{14}{n_m}\log\frac{2}{\delta} \right) \right). \tag{68}$$

Inspired by Juditsky et al. [2011, §4.3.1], we use the value of $p_i$ in (32) to simplify (68). It is easy to verify that

$$\frac{p_{\max}}{p_i} = \frac{1/\sqrt{n_m} + \sqrt{n_m/n_i}}{1/\sqrt{n_m} + \sqrt{n_m/n_1}} \le \left( 1 + \frac{n_m}{\sqrt{n_i}} \right),$$

$$\frac{1}{p_i}\frac{\widetilde{L}}{n_m} = O\left( \frac{p_{\max}}{p_i}\frac{\sqrt{\ln m}}{n_m} \right) = O\left( \left( \frac{1}{n_m} + \frac{1}{\sqrt{n_i}} \right)\sqrt{\ln m} \right), \tag{69}$$

$$p_i \le \left( \frac{1}{\sqrt{n_m}} + 1 \right)\sqrt{\frac{n_i}{n_m}}, \quad \omega_{\max} = \max_{i \in [m]} \frac{p_i^2 n_m}{n_i} \le \left( \frac{1}{\sqrt{n_m}} + 1 \right)^2,$$

$$\frac{1}{p_i}\sqrt{\omega_{\max}} = \frac{1/\sqrt{n_m} + \sqrt{n_m/n_i}}{1/\sqrt{n_m} + 1}\sqrt{\omega_{\max}} \le \frac{1}{\sqrt{n_m}} + \sqrt{\frac{n_m}{n_i}},$$

$$\frac{1}{p_i}\sqrt{\frac{\sigma^2}{n_m}} = O\left( \frac{1}{p_i}\sqrt{\frac{\omega_{\max}(\kappa + \ln^2 m)}{n_m}} \right) = O\left( \left( \frac{1}{n_m} + \frac{1}{\sqrt{n_i}} \right)\sqrt{\kappa + \ln^2 m} \right). \tag{70}$$

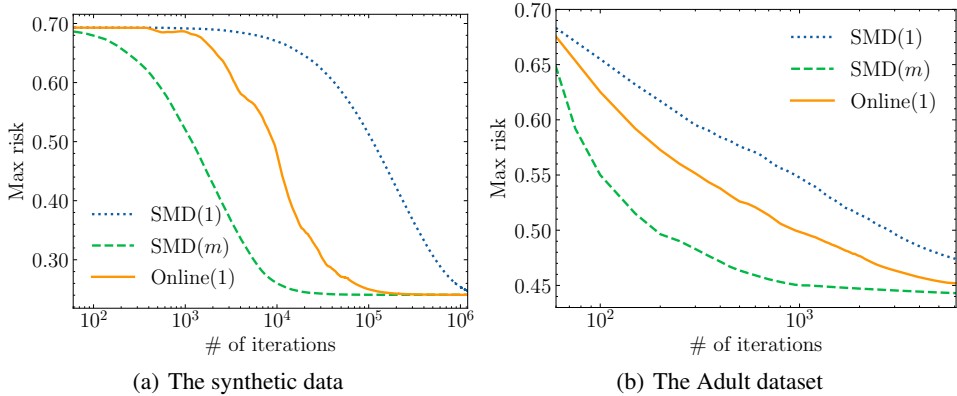

(a) The synthetic data

(b) The Adult dataset

Figure 1: Balanced settings: max risk of different methods versus the number of iterations

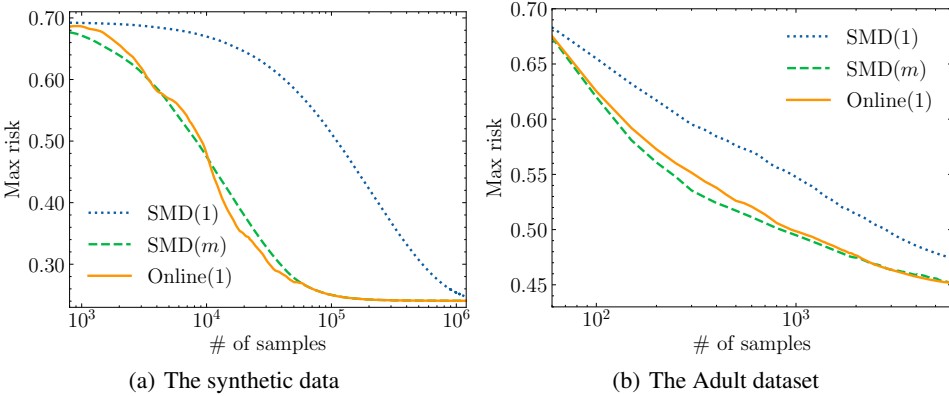

(a) The synthetic data

(b) The Adult dataset

Figure 2: Balanced settings: max risk of different methods versus the number of samples

Substituting (69) and (70) into (68), we have

$$R_i(\bar{\mathbf{w}}) - \frac{1}{p_i} \min_{\mathbf{w} \in \mathcal{W}} \max_{i \in [m]} p_i R_i(\mathbf{w}) = O\left(\left(\frac{1}{n_m} + \frac{1}{\sqrt{n_i}}\right)\sqrt{\kappa + \ln^2 m}\right).$$

## C Experiments

In this section, we conduct empirical studies to evaluate our proposed algorithms.

### C.1 Datasets

Following the setup in previous work [Namkoong and Duchi, 2016, Soma et al., 2022], we use both synthetic and real-world datasets. First, we construct a synthetic data with group number $m = 20$. For each group $i \in [m]$, we draw a model $\mathbf{w}_i^* \in \mathbb{R}^{1000}$ from the uniform distribution over the unit sphere. For distribution $\mathcal{P}_i$, the sample $(\mathbf{x}, y)$ is generated with $\mathbf{x} \sim \mathcal{N}(0, I)$ and $y = \text{sign}(\mathbf{x}^\top \mathbf{w}_i^*)$ with probability $0.9$ and $y = -\text{sign}(\mathbf{x}^\top \mathbf{w}_i^*)$ with probability $0.1$.

We also use the Adult dataset [Becker and Kohavi, 1996], which includes attributes such as age, gender, race, and educational background of $48842$ individuals. The objective is to determine whether an individual's income exceeds $50000$ USD. We set up $m = 6$ groups based on the race and gender attributes, where each group represents a combination of {black, white, others} with {female, male}.

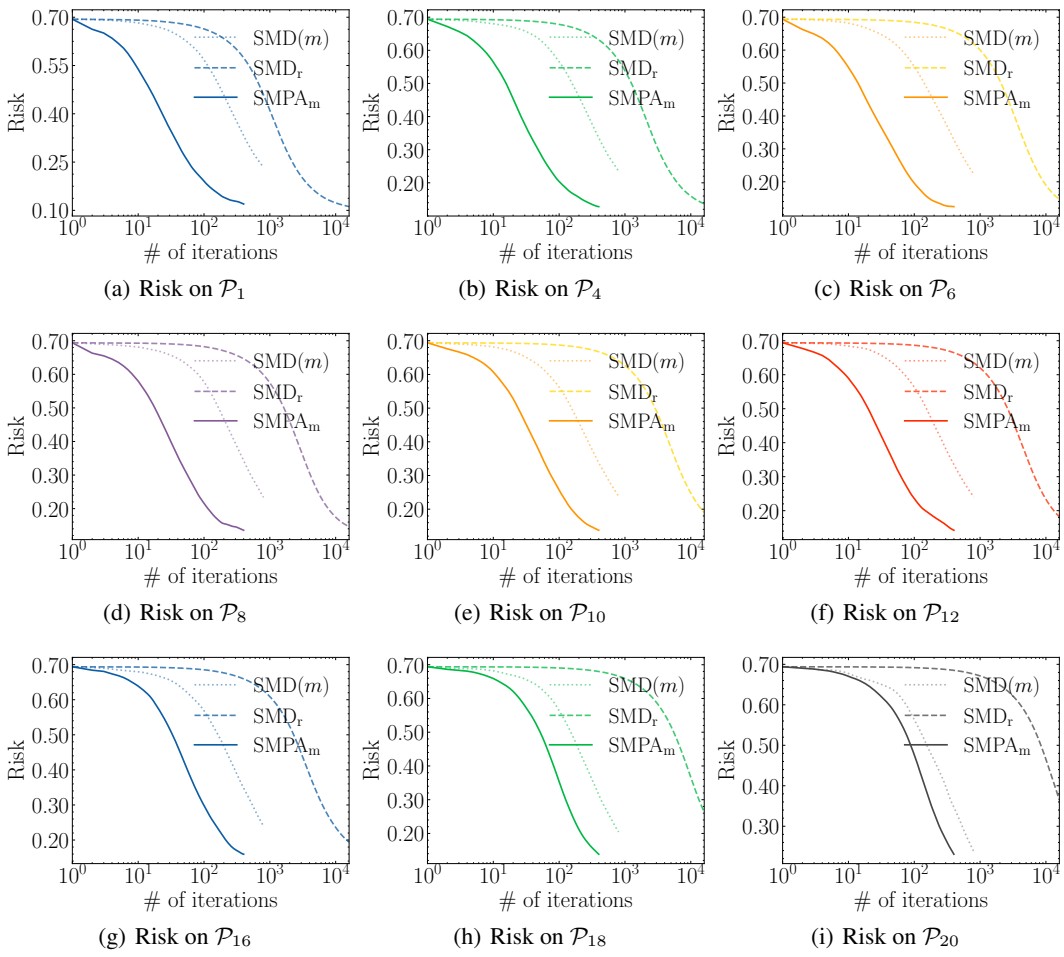

Figure 3: Imbalanced settings with the synthetic dataset: individual risk versus the number of iterations

## C.2 GDRO with Balanced Data

For experiments on the synthetic dataset, we will generate the random sample on the fly, according to the protocol above. For those on the Adult dataset, we will randomly select samples from ech group. In other words, $\mathcal{P}_i$ is defined as the empirical distribution over the data in the $i$-th group.

We refer to the method of Sagawa et al. [2020] and our Algorithm 1 by SMD(1) and SMD($m$) to underscore that they are instances of SMD with 1 sample and $m$ samples in each iteration, respectively. We denote our Algorithm 2 as Online(1) to emphasize that it is based on techniques from online learning and uses 1 sample per iteration. We set $\ell(\cdot;\cdot)$ to be the logistic loss and utilize different methods to train a linear model. In the balanced setting, we use the max risk, i.e., $\max_{i\in[m]} R_i(\mathbf{w})$, as the performance measure. To estimate the risk value, we will draw a substantial number of samples, and use the empirical average to approximate the expectation.

We first report the max risk with respect to the number of iterations in Fig. 1. We observe that SMD($m$) is faster than Online(1), which in turn outperforms SMD(1). This observation is consistent with our theories, since their convergence rates are $O(\sqrt{(\log m)/T})$, $O(\sqrt{m(\log m)/T})$, and $O(m\sqrt{(\log m)/T})$, respectively. Next, we plot the max risk against the number of samples consumed by each algorithm in Fig. 2. As can be seen, the curves of SMD($m$) and Online(1) are very close, indicating that they share the same sample complexity, i.e., $O(m(\log m)/\epsilon^2)$. By contrast, SMD(1) has a much higher sample complexity, i.e., $O(m^2(\log m)/\epsilon^2)$.

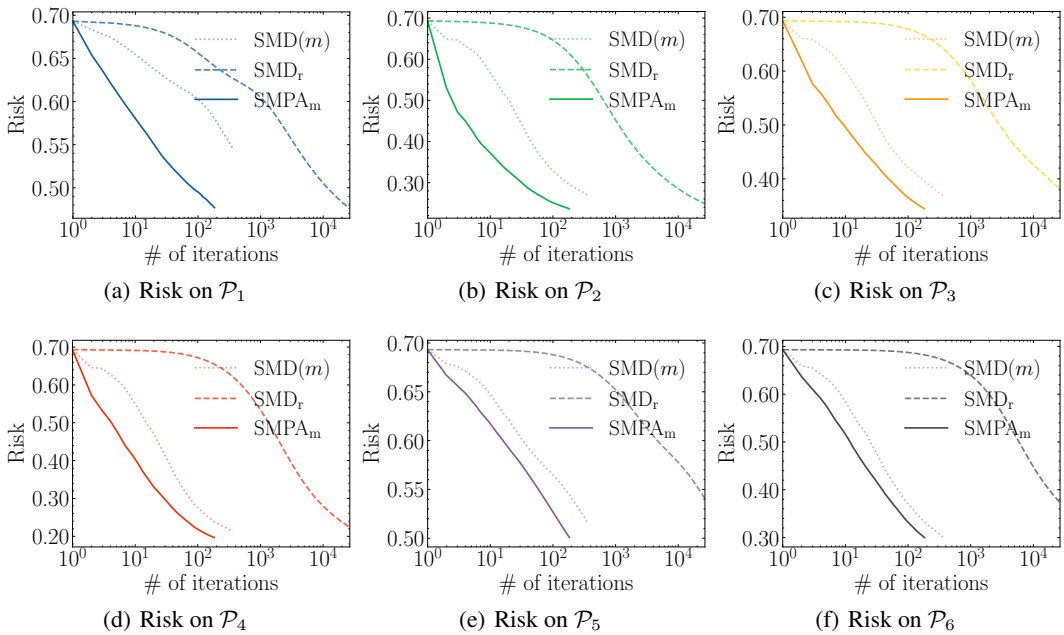

Figure 4: Imbalanced settings with the Adult dataset: individual risk versus the number of iterations

### C.3    GDRO with Imbalanced Data

For experiments on the synthetic dataset, we set the number of samples as $n_i = 800 \times (21 - i)$, and generate each sample as before. For those on the Adult dataset, we first select 364 samples randomly from each group, reserving them for later use in estimating the risk of each group. Then, we visit the remaining samples in each group *once* to simulate the imbalanced setting, where the numbers of samples in 6 groups are 26656, 11518, 1780, 1720, 998, and 364.

Similarly, we label the Baseline mentioned in the first paragraph of Section 3 as $SMD(m)$. We designate our Algorithms 3 and 4 as $SMD_r$ and $SMPA_m$ to highlight that the former combines SMD with random sampling and the latter one integrates SMPA and mini-batches. In the imbalanced setting, we will examine how the risk on each individual distribution decreases with respect to the number of iterations. Recall that the total number of iterations of $SMD(m)$, $SMD_r$ and $SMPA_m$ are $n_m$, $n_1$ and $n_m/2$, respectively.

We present the risk on the individual distribution in Fig. 3 and Fig. 4. First, we observe that our $SMPA_m$ is faster than both $SMD(m)$ and $SMD_r$ across all distributions, and finally attains the lower risk is most cases. This behavior aligns with our Theorem 4, which reveals that $SMPA_m$ achieves a nearly optimal rate of $O((\log m)/\sqrt{n_i})$ for all distributions $\mathcal{P}_i$, after $n_m/2$ iterations. We also note that on distribution $\mathcal{P}_1$, although $SMD_r$ converges slowly, its final risk is the lowest, as illustrated in Fig. 3(a) and Fig. 4(a). This phenomenon is again in accordance with our Theorem 3, which shows that the risk of $SMD_r$ on $\mathcal{P}_1$ reduces at a nearly optimal $O(\sqrt{(\log m)/n_1})$ rate, after $n_1$ iterations. From Fig. 3(i) and Fig. 4(f), we can see that the final risk of $SMD(m)$ on the last distribution $\mathcal{P}_m$ matches that of $SMPA_m$. This outcome is anticipated, as they exhibit similar convergence rates of $O(\sqrt{(\log m)/n_m})$ and $O((\log m)/\sqrt{n_m})$, respectively.

## D    Supporting Lemmas

### D.1    Proof of Lemma 2

The proof follows the argument of Neu [2015, Proof of Lemma 1], and we generalize it to the setting with stochastic rewards. First, observe that for any $i \in [m]$ and $t \in [T]$,

$$\tilde{\xi}_{t,i} = \frac{\hat{\xi}_{t,i}}{p_{t,i} + \gamma_t} \cdot \mathbb{I}[i_t = i]$$

$$\leq \frac{\hat{\xi}_{t,i}}{p_{t,i} + \gamma_t\hat{\xi}_{t,i}} \cdot \mathbb{I}[i_t = i] \qquad\qquad (\hat{\xi}_{t,i} \in [0,1])$$

$$= \frac{1}{2\gamma_t} \frac{2\gamma_t \cdot \hat{\xi}_{t,i}/p_{t,i}}{1 + \gamma_t \cdot \hat{\xi}_{t,i}/p_{t,i}} \cdot \mathbb{I}[i_t = i]$$

$$\leq \frac{1}{\beta_t} \log\left(1 + \beta_t\bar{\xi}_{t,i}\right) \qquad\qquad\qquad (71)$$

where the last step is due to the inequality $\frac{z}{1+z/2} \leq \log(1+z)$ for $z \geq 0$ and we introduce the notations $\beta_t = 2\gamma_t$ and $\bar{\xi}_{t,i} = (\hat{\xi}_{t,i}/p_{t,i}) \cdot \mathbb{I}[i_t = i]$ to simplify the presentation.

Define the notation $\tilde{\lambda}_t = \sum_{i=1}^{m} \alpha_{t,i}\tilde{\xi}_{t,i}$ and $\lambda_t = \sum_{i=1}^{m} \alpha_{t,i}\xi_{t,i}$. Then, we have

$$\mathrm{E}_{t-1}\left[\exp(\tilde{\lambda}_t)\right] = \mathrm{E}_{t-1}\left[\exp\left(\sum_{i=1}^{m} \alpha_{t,i}\tilde{\xi}_{t,i}\right)\right]$$

$$\overset{(71)}{\leq} \mathrm{E}_{t-1}\left[\exp\left(\sum_{i=1}^{m} \frac{\alpha_{t,i}}{\beta_t} \log\left(1 + \beta_t\bar{\xi}_{t,i}\right)\right)\right]$$

$$\leq \mathrm{E}_{t-1}\left[\exp\left(\sum_{i=1}^{m} \log\left(1 + \alpha_{t,i}\bar{\xi}_{t,i}\right)\right)\right] \qquad (\tfrac{\alpha_{t,i}}{\beta_t} \leq 1 \text{ by assumption})$$

$$= \mathrm{E}_{t-1}\left[\Pi_{i=1}^{m}\left(1 + \alpha_{t,i}\bar{\xi}_{t,i}\right)\right]$$

$$= \mathrm{E}_{t-1}\left[1 + \sum_{i=1}^{m} \alpha_{t,i}\bar{\xi}_{t,i}\right]$$

$$= 1 + \sum_{i=1}^{m} \alpha_{t,i}\xi_{t,i} \leq \exp\left(\sum_{i=1}^{m} \alpha_{t,i}\xi_{t,i}\right) = \exp(\lambda_t) \qquad (72)$$

where the second inequality is by the inequality $x\log(1+y) \leq \log(1+xy)$ that holds for all $y \geq -1$ and $x \in [0,1]$, and the equality $\mathrm{E}_{t-1}\left[\Pi_{i=1}^{m}\left(1 + \alpha_{t,i}\bar{\xi}_{t,i}\right)\right] = \mathrm{E}_{t-1}\left[1 + \sum_{i=1}^{m} \alpha_{t,i}\bar{\xi}_{t,i}\right]$ follows from the fact that $\bar{\xi}_{t,i} \cdot \bar{\xi}_{t,j} = 0$ holds whenever $i \neq j$. The last line is due to $\mathrm{E}_{t-1}[\bar{\xi}_{t,i}] = \mathrm{E}_{t-1}[(\hat{\xi}_{t,i}/p_{t,i}) \cdot \mathbb{I}[i_t = i]] = \xi_{t,i}$ and the inequality $1 + z \leq e^z$ for all $z \in \mathbb{R}$.

As a result, from (72) we conclude that the process $Z_t = \exp\left(\sum_{s=1}^{t}(\tilde{\lambda}_s - \lambda_s)\right)$ is a supermartingale. Indeed, $\mathrm{E}_{t-1}[Z_t] = \mathrm{E}_{t-1}\left[\exp\left(\sum_{s=1}^{t-1}(\tilde{\lambda}_s - \lambda_s)\right) \cdot \exp(\tilde{\lambda}_t - \lambda_t)\right] \leq Z_{t-1}$. Thus, we have $\mathrm{E}[Z_T] \leq \mathrm{E}[Z_{T-1} \leq \ldots \leq \mathrm{E}[Z_0] = 1$. By Markov's inequality,

$$\Pr\left[\sum_{t=1}^{T}(\tilde{\lambda}_t - \lambda_t) > \epsilon\right] \leq \mathrm{E}\left[\exp\left(\sum_{t=1}^{T}(\tilde{\lambda}_t - \lambda_t)\right)\right] \cdot \exp(-\epsilon) \leq \exp(-\epsilon)$$

holds for any $\epsilon > 0$. By setting $\exp(-\epsilon) = \delta$ and solving the value, we complete the proof for (54). And the inequality (55) for the scenario $\gamma_t = \gamma$ can be immediately obtained by setting $\alpha_{t,i} = 2\gamma$ and taking the union bound over all $i \in [m]$.

## D.2 Proof of Lemma 3

From the definition of norms in (34), we have

$$\|F([\mathbf{w};\mathbf{q}]) - F([\mathbf{w}';\mathbf{q}'])\|_*^2$$

$$= \left\| \left[ \sum_{i=1}^m q_i p_i \nabla R_i(\mathbf{w}) - \sum_{i=1}^m q_i' p_i \nabla R_i(\mathbf{w}'); \left[p_1 R_1(\mathbf{w}') - p_1 R_1(\mathbf{w}), \ldots, p_m R_m(\mathbf{w}') - p_m R_m(\mathbf{w})\right]^\top \right] \right\|_*^2$$

$$= 2D^2 \left\| \sum_{i=1}^m q_i p_i \nabla R_i(\mathbf{w}) - \sum_{i=1}^m q_i' p_i \nabla R_i(\mathbf{w}') \right\|_{w,*}^2$$

$$+ 2 \left\| \left[p_1 R_1(\mathbf{w}') - p_1 R_1(\mathbf{w}), \ldots, p_m R_m(\mathbf{w}') - p_m R_m(\mathbf{w})\right]^\top \right\|_\infty^2 \ln m$$

$$= 2D^2 \left\| \sum_{i=1}^m q_i p_i \nabla R_i(\mathbf{w}) - \sum_{i=1}^m q_i' p_i \nabla R_i(\mathbf{w}) + \sum_{i=1}^m q_i' p_i \nabla R_i(\mathbf{w}) - \sum_{i=1}^m q_i' p_i \nabla R_i(\mathbf{w}') \right\|_{w,*}^2$$

$$+ 2 \left\| \left[p_1 R_1(\mathbf{w}') - p_1 R_1(\mathbf{w}), \ldots, p_m R_m(\mathbf{w}') - p_m R_m(\mathbf{w})\right]^\top \right\|_\infty^2 \ln m$$

$$\leq 4D^2 \underbrace{\left\| \sum_{i=1}^m q_i p_i \nabla R_i(\mathbf{w}) - \sum_{i=1}^m q_i' p_i \nabla R_i(\mathbf{w}) \right\|_{w,*}^2}_{:=A} + 4D^2 \underbrace{\left\| \sum_{i=1}^m q_i' p_i \nabla R_i(\mathbf{w}) - \sum_{i=1}^m q_i' p_i \nabla R_i(\mathbf{w}') \right\|_{w,*}^2}_{:=B}$$

$$+ \underbrace{2 \max_{i \in [m]} \left| p_i \left[ R_i(\mathbf{w}) - R_i(\mathbf{w}') \right] \right|^2 \ln m}_{:=C} .$$

To bound term $A$, we have

$$4D^2 \left\| \sum_{i=1}^m q_i p_i \nabla R_i(\mathbf{w}) - \sum_{i=1}^m q_i' p_i \nabla R_i(\mathbf{w}) \right\|_{w,*}^2$$

$$\leq 4D^2 \left( \sum_{i=1}^m |q_i - q_i'| \|p_i \nabla R_i(\mathbf{w})\|_{w,*} \right)^2 \overset{(60)}{\leq} 4D^2 \left( \sum_{i=1}^m |q_i - q_i'| p_i G \right)^2 \leq 4D^2 G^2 p_{\max}^2 \|\mathbf{q} - \mathbf{q}'\|_1^2 .$$

where $p_{\max}$ is defined in (31). To bound $B$, we have

$$4D^2 \left\| \sum_{i=1}^m q_i' p_i \nabla R_i(\mathbf{w}) - \sum_{i=1}^m q_i' p_i \nabla R_i(\mathbf{w}') \right\|_{w,*}^2$$

$$\leq 4D^2 \left( \sum_{i=1}^m q_i' p_i \|\nabla R_i(\mathbf{w}) - \nabla R_i(\mathbf{w}')\|_{w,*} \right)^2 \overset{(30)}{\leq} 4D^2 \left( \sum_{i=1}^m q_i' p_i L \|\mathbf{w} - \mathbf{w}'\|_w \right)^2$$

$$\leq 4D^2 L^2 p_{\max}^2 \|\mathbf{w} - \mathbf{w}'\|_w^2 \left( \sum_{i=1}^m q_i' \right)^2 = 4D^2 L^2 p_{\max}^2 \|\mathbf{w} - \mathbf{w}'\|_w^2 .$$

To bound $C$, we have

$$2 \max_{i \in [m]} \left| p_i \left[ R_i(\mathbf{w}) - R_i(\mathbf{w}') \right] \right|^2 \ln m$$

$$\overset{(61)}{\leq} 2 \max_{i \in [m]} |p_i G \|\mathbf{w} - \mathbf{w}'\|_w|^2 \ln m \leq 2 G^2 p_{\max}^2 \|\mathbf{w} - \mathbf{w}'\|_w^2 \ln m .$$

Putting everything together, we have

$$\|F([\mathbf{w};\mathbf{q}]) - F([\mathbf{w}';\mathbf{q}'])\|_*^2 \leq (4D^2 L^2 + 2G^2 \ln m) p_{\max}^2 \|\mathbf{w} - \mathbf{w}'\|_w^2 + 4D^2 G^2 p_{\max}^2 \|\mathbf{q} - \mathbf{q}'\|_1^2$$

$$\leq p_{\max}^2 (8D^4 L^2 + 8D^2 G^2 \ln m) \left( \frac{1}{2D^2} \|\mathbf{w} - \mathbf{w}'\|_w^2 + \frac{1}{2 \ln m} \|\mathbf{q} - \mathbf{q}'\|_1^2 \right)$$

$$= p_{\max}^2 (8D^4 L^2 + 8D^2 G^2 \ln m) \left\| [\mathbf{w} - \mathbf{w}';\mathbf{q} - \mathbf{q}'] \right\|^2$$

which implies

$$\|F([\mathbf{w};\mathbf{q}]) - F([\mathbf{w}';\mathbf{q}'])\|_* \le p_{\max}\sqrt{8D^4L^2 + 8D^2G^2\ln m}\,\big\|[\mathbf{w}-\mathbf{w}';\mathbf{q}-\mathbf{q}']\big\| \le \widetilde{L}\big\|[\mathbf{w}-\mathbf{w}';\mathbf{q}-\mathbf{q}']\big\|$$

where $\widetilde{L}$ is defined in (31).

### D.3 Proof of Lemma 4

The light tail condition, required by Juditsky et al. [2011], is essentially the sub-Gaussian condition. To this end, we introduce the following sub-gaussian properties [Vershynin, 2018, Proposition 2.5.2].

**Proposition 1 (Sub-gaussian properties)** *Let $X$ be a random variable. Then the following properties are equivalent; the parameters $K_i > 0$ appearing in these properties differ from each other by at most an absolute constant factor.*

*(i) The tails of $X$ satisfy*

$$\Pr[|X| \ge t] \le 2\exp(-t^2/K_1^2),\ \forall t \ge 0.$$

*(ii) The moments of $X$ satisfy*

$$\|X\|_{L_p} = (\mathrm{E}|X|^p)^{1/p} \le K_2\sqrt{p},\ \forall p \ge 1.$$

*(iii) The moment generating function (MGF) of $X^2$ satisfies*

$$\mathrm{E}\big[\exp(\lambda^2 X^2)\big] \le \exp(K_3^2\lambda^2),\ \forall \lambda \text{ such that } |\lambda| \le 1/K_3.$$

*(iv) The MGF of $X^2$ is bounded at some point, namely*

$$\mathrm{E}\big[\exp(X^2/K_4^2)\big] \le 2.$$

*Moreover, if $\mathrm{E}[X] = 0$ then properties (i)–(iv) are also equivalent to the following property.*
*(v) The MGF of X satisfies*

$$\mathrm{E}\big[\exp(\lambda X)\big] \le \exp(K_5^2\lambda^2),\ \forall \lambda \in \mathbb{R}.$$

From the above proposition, we observe that the exact value of those constant $K_1, \ldots, K_5$ is not important, and it is very tedious to calculate them. So, in the following, we only focus on the order of those constants. To simplify presentations, we use $c$ to denote an absolute constant that is independent of all the essential parameters, and its value may change from line to line.

Since

$$\begin{aligned}
&\|F([\mathbf{w};\mathbf{q}]) - \mathbf{g}([\mathbf{w};\mathbf{q}])\|_*^2 \\
&= 2D^2\|\nabla_{\mathbf{w}}\varphi(\mathbf{w},\mathbf{q}) - \mathbf{g}_w(\mathbf{w},\mathbf{q})\|_{w,*}^2 + 2\|\nabla_{\mathbf{q}}\varphi(\mathbf{w},\mathbf{q}) - \mathbf{g}_q(\mathbf{w},\mathbf{q})\|_\infty^2 \ln m,
\end{aligned}$$

we proceed to analyze the behavior of $\|\nabla_{\mathbf{w}}\varphi(\mathbf{w},\mathbf{q}) - \mathbf{g}_w(\mathbf{w},\mathbf{q})\|_{w,*}^2$ and $\|\nabla_{\mathbf{q}}\varphi(\mathbf{w},\mathbf{q}) - \mathbf{g}_q(\mathbf{w},\mathbf{q})\|_\infty^2$. To this end, we have the following lemma.

**Lemma 5** *We have*

$$\begin{aligned}
\mathrm{E}\left[\exp\left(\frac{1}{c\kappa G^2\omega_{\max}}\|\nabla_{\mathbf{w}}\varphi(\mathbf{w},\mathbf{q}) - \mathbf{g}_w(\mathbf{w},\mathbf{q})\|_{w,*}^2\right)\right] &\le 2, \\
\mathrm{E}\left[\exp\left(\frac{1}{c\omega_{\max}\ln m}\|\nabla_{\mathbf{q}}\varphi(\mathbf{w},\mathbf{q}) - \mathbf{g}_q(\mathbf{w},\mathbf{q})\|_\infty^2\right)\right] &\le 2
\end{aligned} \tag{73}$$

*where $\omega_{\max}$ is defined in (31) and $c > 0$ is an absolute constant.*

From Lemma 5, we have

$$\mathrm{E}\left[\exp\left(\frac{1}{2c\kappa D^2 G^2 \omega_{\max} + 2c\omega_{\max}\ln^2 m}\|F([\mathbf{w};\mathbf{q}]) - \mathbf{g}([\mathbf{w},\mathbf{q}])\|_*^2\right)\right]$$

$$=\mathrm{E}\left[\exp\left(\frac{2D^2}{2c\kappa D^2 G^2 \omega_{\max} + 2c\omega_{\max}\ln^2 m}\|\nabla_{\mathbf{w}}\varphi(\mathbf{w},\mathbf{q}) - \mathbf{g}_w(\mathbf{w},\mathbf{q})\|_{w,*}^2\right.\right.$$
$$\left.\left.+\frac{2\ln m}{2c\kappa D^2 G^2 \omega_{\max} + 2c\omega_{\max}\ln^2 m}\|\nabla_{\mathbf{q}}\varphi(\mathbf{w},\mathbf{q}) - \mathbf{g}_q(\mathbf{w},\mathbf{q})\|_\infty^2\right)\right]$$

$$=\mathrm{E}\left[\exp\left(\frac{\kappa D^2 G^2}{\kappa D^2 G^2 + \ln^2 m}\frac{\|\nabla_{\mathbf{w}}\varphi(\mathbf{w},\mathbf{q}) - \mathbf{g}_w(\mathbf{w},\mathbf{q})\|_{w,*}^2}{c\kappa G^2 \omega_{\max}}\right.\right.$$
$$\left.\left.+\frac{\ln^2 m}{\kappa D^2 G^2 + \ln^2 m}\frac{\|\nabla_{\mathbf{q}}\varphi(\mathbf{w},\mathbf{q}) - \mathbf{g}_q(\mathbf{w},\mathbf{q})\|_\infty^2}{c\omega_{\max}\ln m}\right)\right]$$

$$\leq\frac{\kappa D^2 G^2}{\kappa D^2 G^2 + \ln^2 m}\mathrm{E}\left[\exp\left(\frac{\|\nabla_{\mathbf{w}}\varphi(\mathbf{w},\mathbf{q}) - \mathbf{g}_w(\mathbf{w},\mathbf{q})\|_{w,*}^2}{c\kappa G^2 \omega_{\max}}\right)\right]$$
$$+\frac{\ln^2 m}{\kappa D^2 G^2 + \ln^2 m}\mathrm{E}\left[\exp\left(\frac{\|\nabla_{\mathbf{q}}\varphi(\mathbf{w},\mathbf{q}) - \mathbf{g}_q(\mathbf{w},\mathbf{q})\|_\infty^2}{c\omega_{\max}\ln m}\right)\right]$$

$$\overset{(73)}{\leq}\frac{\kappa D^2 G^2}{\kappa D^2 G^2 + \ln^2 m}2 + \frac{\ln^2 m}{\kappa D^2 G^2 + \ln^2 m}2 = 2$$

where the first inequality follows from Jensen's inequality.

### D.4 Proof of Lemma 5

To analyze $\|\nabla_{\mathbf{w}}\varphi(\mathbf{w},\mathbf{q}) - \mathbf{g}_w(\mathbf{w},\mathbf{q})\|_{w,*}^2$, we first consider the approximation error caused by samples from $\mathcal{P}_i$:

$$\left\|\frac{n_m}{n_i}\sum_{j=1}^{n_i/n_m}\nabla\ell(\mathbf{w};\mathbf{z}^{(i,j)}) - \nabla R_i(\mathbf{w})\right\|_{w,*} = \left\|\frac{n_m}{n_i}\sum_{j=1}^{n_i/n_m}\left[\nabla\ell(\mathbf{w};\mathbf{z}^{(i,j)}) - \nabla R_i(\mathbf{w})\right]\right\|_{w,*}.$$

Under the regularity condition of $\|\cdot\|_{w,*}$ in Assumption 6, we have, for any $\gamma \geq 0$,

$$\left\|\frac{n_m}{n_i}\sum_{j=1}^{n_i/n_m}\left[\nabla\ell(\mathbf{w};\mathbf{z}^{(i,j)}) - \nabla R_i(\mathbf{w})\right]\right\|_{w,*} \geq 2G(\sqrt{2\kappa} + \sqrt{2}\gamma)\sqrt{\frac{n_m}{n_i}} \leq \exp(-\gamma^2/2) \quad (74)$$

which is a directly consequence of the concentration inequality of vector norms [Juditsky and Nemirovski, 2008, Theorem 2.1.(iii)] and (62). Then, we introduce the following lemma to simplify (74).

**Lemma 6** *Suppose we have*

$$\Pr\left[X \geq \alpha + \gamma\right] \leq \exp(-\gamma^2/2),\ \forall\gamma > 0$$

*where $X$ is nonnegative. Then, we have*

$$\Pr\left[X \geq \gamma\right] \leq 2\exp\left(-\gamma^2/\max(6\alpha^2, 8)\right),\ \forall\gamma > 0.$$

From (74) and Lemma 6, we have

$$\Pr\left[\frac{1}{2\sqrt{2}G}\sqrt{\frac{n_i}{n_m}}\left\|\frac{n_m}{n_i}\sum_{j=1}^{n_i/n_m}\left[\nabla\ell(\mathbf{w};\mathbf{z}^{(i,j)}) - \nabla R_i(\mathbf{w})\right]\right\|_{w,*} \geq \gamma\right]$$

$$\leq 2\exp\left(-\gamma^2/\max(6\kappa, 8)\right) \leq 2\exp\left(-\gamma^2/(8\kappa)\right),\ \forall\gamma > 0$$

which satisfies the Proposition 1.(i). From the equivalence between Proposition 1.(i) and Proposition 1.(iv), we have

$$
\mathrm{E}\left[\exp\left(\left\|\frac{n_m}{n_i}\sum_{j=1}^{n_i/n_m}\left[\nabla\ell(\mathbf{w};\mathbf{z}^{(i,j)})-\nabla R_i(\mathbf{w})\right]\right\|_{w,*}^2\bigg/\frac{c\kappa G^2 n_m}{n_i}\right)\right]\leq 2.
$$

Inserting the scaling factor $p_i$, we have

$$
\mathrm{E}\left[\exp\left(\left\|p_i\frac{n_m}{n_i}\sum_{j=1}^{n_i/n_m}\left[\nabla\ell(\mathbf{w};\mathbf{z}^{(i,j)})-\nabla R_i(\mathbf{w})\right]\right\|_{w,*}^2\bigg/\frac{c\kappa G^2 p_i^2 n_m}{n_i}\right)\right]\leq 2. \tag{75}
$$

To simplify the notation, we define

$$
\mathbf{u}_i=p_i\frac{n_m}{n_i}\sum_{j=1}^{n_i/n_m}\left[\nabla\ell(\mathbf{w};\mathbf{z}^{(i,j)})-\nabla R_i(\mathbf{w})\right],\text{ and }\omega_{\max}=\max_{i\in[m]}\frac{p_i^2 n_m}{n_i}.
$$

By Jensen's inequality, we have

$$
\mathrm{E}\left[\exp\left(\frac{1}{c\kappa G^2\omega_{\max}}\|\nabla_{\mathbf{w}}\varphi(\mathbf{w},\mathbf{q})-\mathbf{g}_w(\mathbf{w},\mathbf{q})\|_{w,*}^2\right)\right]
$$

$$
=\mathrm{E}\left[\exp\left(\left\|\sum_{i=1}^m q_i\mathbf{u}_i\right\|_{w,*}^2\bigg/\left[c\kappa G^2\omega_{\max}\right]\right)\right]
$$

$$
\leq\sum_{i=1}^m q_i\mathrm{E}\left[\exp\left(\|\mathbf{u}_i\|_{w,*}^2\bigg/\left[c\kappa G^2\omega_{\max}\right]\right)\right]\stackrel{(75)}{\leq}\sum_{i=1}^m q_i 2=2.
$$

where we use the fact that $\|\cdot\|_{w,*}$, $(\cdot)^2$ and $\exp(\cdot)$ are convex, and the last two functions are increasing in $\mathbb{R}_+$.

To analyze $\|\nabla_{\mathbf{q}}\varphi(\mathbf{w},\mathbf{q})-\mathbf{g}_q(\mathbf{w},\mathbf{q})\|_\infty^2$, we consider the approximation error related to $\mathcal{P}_i$:

$$
\left|\frac{n_m}{n_i}\sum_{j=1}^{n_i/n_m}\ell(\mathbf{w};\mathbf{z}^{(i,j)})-R_i(\mathbf{w})\right|=\left|\frac{n_m}{n_i}\sum_{j=1}^{n_i/n_m}\left[\ell(\mathbf{w};\mathbf{z}^{(i,j)})-R_i(\mathbf{w})\right]\right|.
$$

Note that the absolute value $|\cdot|$ is 1-regular [Juditsky and Nemirovski, 2008]. Following (59) and the derivation of (75), we have

$$
\mathrm{E}\left[\exp\left(\left|p_i\frac{n_m}{n_i}\sum_{j=1}^{n_i/n_m}\left[\ell(\mathbf{w};\mathbf{z}^{(i,j)})-R_i(\mathbf{w})\right]\right|^2\bigg/\frac{cp_i^2 n_m}{n_i}\right)\right]\leq 2. \tag{76}
$$

To prove that $\|\nabla_{\mathbf{q}}\varphi(\mathbf{w},\mathbf{q})-\mathbf{g}_q(\mathbf{w},\mathbf{q})\|_\infty^2$ is also sub-Gaussian, we need to analyze the effect of the infinity norm. To this end, we develop the following lemma.

**Lemma 7** *Suppose*

$$
\mathrm{E}\left[\exp\left(|X_j|^2/K_j^2\right)\right]\leq 2,\ \forall j\in[m]. \tag{77}
$$

*Then,*

$$
\mathrm{E}\left[\exp\left(\max_{j\in[m]}|X_j|^2\bigg/\left[cK_{\max}^2\ln m\right]\right)\right]\leq 2.
$$

*where $c>0$ is an absolute constant, and $K_{\max}=\max_{j\in[m]}K_j$.*

From (76) and Lemma 7, we have

$$
\mathrm{E}\left[\exp\left(\frac{1}{c\omega_{\max}\ln m}\|\nabla_{\mathbf{q}}\varphi(\mathbf{w},\mathbf{q})-\mathbf{g}_q(\mathbf{w},\mathbf{q})\|_\infty^2\right)\right]\leq 2.
$$

### D.5  Proof of Lemma 6

When $\gamma \in [0, 2\alpha]$, we have

$$\Pr[X \geq \gamma] \leq 1 \leq 2\exp(-2/3) \leq 2\exp(-\gamma^2/6\alpha^2).$$

When $\gamma \geq 2\alpha$, we have

$$\Pr[X \geq \gamma] = \Pr[X \geq \alpha + \gamma - \alpha] \leq \exp(-(\gamma-\alpha)^2/2) \leq \exp(-\gamma^2/8)$$

where we use the fact

$$\gamma - \alpha \geq \frac{\gamma}{2}.$$

Thus, we always have

$$\Pr[X \geq \gamma] \leq 2\exp\left(-\gamma^2/\max(6\alpha^2, 8)\right), \ \forall \gamma > 0.$$

### D.6  Proof of Lemma 7

From (77), and the equivalence between Proposition 1.(i) and Proposition 1.(iv), we have

$$\Pr\left[|X_j| \geq t\right] \leq 2\exp\left(-t^2/cK_j^2\right), \ \forall t \geq 0, \forall j \in [m].$$

As a result,

$$\Pr\left[\max_{j\in[m]}|X_j| \geq t\right] = \Pr\left[\exists j, |X_j| \geq t\right] \leq \sum_{j=1}^{m}\Pr\left[|X_j| \geq t\right] \leq 2\sum_{j=1}^{m}\exp\left(-t^2/cK_j^2\right)$$

$$\leq 2m\exp\left(-t^2/cK_{\max}^2\right) = \exp\left(-t^2/cK_{\max}^2 + \ln[2m]\right).$$

Choosing $t = \sqrt{cK_{\max}^2(\ln[2m] + \gamma^2/2)}$, we have

$$\Pr\left[\max_{j\in[m]}|X_j| \geq \sqrt{cK_{\max}^2(\ln[2m] + \gamma^2/2)}\right] \leq \exp\left(-\gamma^2/2\right).$$

Thus

$$\Pr\left[\max_{j\in[m]}|X_j| \geq \sqrt{cK_{\max}^2}\left(\sqrt{\ln[2m]} + \gamma/\sqrt{2}\right)\right] \leq \exp\left(-\gamma^2/2\right)$$

$$\Leftrightarrow \Pr\left[\sqrt{\frac{2}{cK_{\max}^2}}\max_{j\in[m]}|X_j| \geq \sqrt{2\ln[2m]} + \gamma\right] \leq \exp\left(-\gamma^2/2\right).$$

By Lemma 6, we have

$$\Pr\left[\sqrt{\frac{2}{cK_{\max}^2}}\max_{j\in[m]}|X_j| \geq \gamma\right] \leq 2\exp\left(-\gamma^2/\max\left(12\cdot\ln[2m], 8\right)\right), \ \forall \gamma > 0.$$

From the equivalence between Proposition 1.(i) and Proposition 1.(iv), we have

$$\mathrm{E}\left[\exp\left(\max_{j\in[m]}|X_j|^2 \middle/ \left[cK_{\max}^2\ln m\right]\right)\right] \leq 2.$$

