# OpenReview forum: "Stochastic Approximation Approaches to Group Distributionally Robust Optimization"
_NeurIPS.cc/2023/Conference — NeurIPS 2023 poster_

### Official Review · Reviewer_JWj5 · 2023-06-20

**Soundness:** 3 good
**Presentation:** 4 excellent
**Contribution:** 4 excellent
**Rating:** 7
**Confidence:** 4

**Summary:**

This paper provides near-optimal optimization algorithm to group distributionally robust optimization problem (GDRO). Since it is a special case of the stochastic convex-concave saddle point algorithm, the standard SMD could indeed achieve near-optimal sample complexity. Besides, the authors also apply online learning techniques to reduce the number of samples required in each iteration. Finally, the authors consider a complicated scenario where the number of samples that can be drawn from each target distribution could be different. Then the authors consider a weighted GDRO formulation and the convergence analysis of optimization algorithm.

**Strengths:**

- Results are presented in a clear way. The proof is also fluent. It should be accepted.
- SMD and the online learning fashion of SMD are used in a cleaver way such that the authors achieved near-optimal sample complexity, and the per-iteration sample complexity cost is reduced (the per-iteraiton complexity is important because it relates to the storage cost of an optimization algorithm).
- The notion of weighted GDRO is also of research interest.

**Weaknesses:**

- No numerical study is presented to demonstrate the convergence rate of proposed algorithm.

**Questions:**

N/A

**Limitations:**

The authors properly discussed relate work regarding optimization algorithms for DRO. I think one reference could be added: Wang et. al, Sinkhorn Distributionally Robust Optimization, arXiv preprint arXiv:2109.11926, June 2023.

Reason: The authors listed some progress on solving Wasserstein DRO. The reference (Wang et. al, 2023) provides a SMD algorithm with biased gradient oracles that solves entropic-regularized DRO problem up to $\delta$-optimality gap with complexity $\tilde{O}(\delta^{-2})$.

---

> ### Author Rebuttal · Authors · 2023-08-05
>
> Many thanks for the constructive reviews!
>
> ---
>
> Q1: No numerical study is presented to demonstrate the convergence rate of proposed algorithm
>
> A1: We have carried out preliminary experiments, the results of which can be found in the global response to all reviewers. These empirical findings align closely with our theories.
>
> ---
> Q2: I think one reference could be added …
>
> A2: Thank you for identifying the missing reference. We will read and cite this paper.

---

> > ### Comment · Reviewer_JWj5 · 2023-08-10
> > **After reading rebuttal**
> >
> > I have read rebuttal, from which I can tell the authors put into lots of efforts. I will keep the score of 7 as it is.

---

> > > ### Author Response · Authors · 2023-08-10
> > >
> > > Dear Reviewer JWj5,
> > >
> > > Thank you very much for your kind reply! We will revise our paper according to the constructive reviews.
> > >
> > > Best
> > >
> > > Authors

---

### Official Review · Reviewer_EKF4 · 2023-07-01

**Soundness:** 3 good
**Presentation:** 3 good
**Contribution:** 2 fair
**Rating:** 5
**Confidence:** 3

**Summary:**

The paper considers group DRO and shows an SMD algorithm to optimize. This algorithm requires a sample from every distribution in each iteration and matches the sample complexity lower bound up to a log factor. The second algorithm is designed for the case when the online functions depend on past decisions. For this, the authors use SMD and Exp3-IX. Finally, for the case with sample budgets from each distribution, the paper proposes two algorithms, one satisfies the constraints on the number of samples in expectation and the other using mini-batches satisfies the exact constraints.

**Strengths:**

- Despite some weaknesses, I think all algorithms proposed in this paper are simple and natural, which at the same time achieve optimal sample complexity.
- The presentation is clear to follow with proper explanations and intuitions.

**Weaknesses:**

1. I think the weakness of the first algorithm is that in many cases, it might be impossible to obtain samples from all distributions.
2. I’m not sure I understand the novelty of the second algorithm compared with Soma, Gatmiry & Jegelka (2022). In the comparison after remark 3, the authors point out an issue with this prior work. The authors use an alternative objective ($\epsilon_\phi$) to control the optimality gap, while Soma, Gatmiry & Jegelka (2022) directly bound the optimality gap instead of $\epsilon_\phi$, I don’t see any issue with not being able to bound $\epsilon_\phi$. Note that in Soma, Gatmiry & Jegelka (2022), the authors also propose a few algorithms in the same spirit as Algorithm 2, which I think work for the non-oblivious case as well.
3.  Mini-batch sizes in Algorithm 4 can be very large.
4. No experiments

**Questions:**

1. Can the authors discuss more the comparison with Soma, Gatmiry & Jegelka (2022) in my concern above?
2. I think it would be better to add some experiments.

**Limitations:**

The authors discuss some limitations of the proposed algorithms.

---

> ### Author Rebuttal · Authors · 2023-08-05
>
> Many thanks for the constructive reviews! We believe there are some misunderstandings about the work of Soma et al. [2022], and provide detailed responses below. We hope the reviewer could examine them, and reassess the significance of our results. We are looking forward to addressing any further question in the reviewer-author discussion period.
>
> ---
>
> Q1: In the comparison after remark 3, the authors point out an issue with this prior work. The authors use an alternative objective ($\epsilon_\phi$) to control the optimality gap, while Soma, Gatmiry & Jegelka (2022) directly bound the optimality gap instead of $\epsilon_\phi$. I don’t see any issue with not being able to bound $\epsilon_\phi$. Note that in Soma, Gatmiry & Jegelka (2022), the authors also propose a few algorithms in the same spirit as Algorithm 2, which I think work for the non-oblivious case as well.
>
> A1: In fact, the analysis of Soma et al. [2022] is *flawed*, and they fail to bound the optimality gap. Besides, their analysis is based on the theoretical guarantee for oblivious online learning [Orabona, 2019], and thus does not work for the non-oblivious case.
>
> Below, we explain why their analysis is problematic. To facilitate understandings, we use the notations of Soma et al. [2022]. From (12) of their paper, we know that the key is to bound the expectation of two regret:
> $$
> \mathbf{E}[R_\theta(T)]= \mathbf{E}\left[ \sum_{t=1}^T L(\theta_t,q_t)\right] - \mathbf{E}\left[\min_{\theta \in \Theta} \sum_{t=1}^T L(\theta,q_t) \right]
> $$
> and
> $$
> \mathbf{E} [R_q(T)]= \mathbf{E}\left[ \max_{q \in \Delta_m} \sum_{t=1}^T L(\theta,q_t) \right]- \mathbf{E}\left[ \sum_{t=1}^T L(\theta_t,q_t) \right].
> $$
> Then, in Section B.1, they make use of Lemmas 2 and 3 to bound $\mathbf{E}[R_\theta(T)] $ and $\mathbf{E} [R_q(T)]$, which is unfortunately incorrect. Let's take $R_\theta(T)$ as an example to elucidate the reason. From Lemma 3, they can bound the regret in terms of the random function $\ell(\theta,z_t)$, where $z_t$ is the random sample generated in the $t$-th round:
> $$
> r_\theta(T)=\sum_{t=1}^T \ell(\theta_t,z_t)- \min_{\theta \in \Theta} \sum_{t=1}^T \ell(\theta,z_t).
> $$
> However, the expectation of $r_\theta(T)$ is *not* equal to $\mathbf{E}[R_\theta(T)]$, and thus Lemma 3 cannot be applied. To see this, we have
> $$
> \mathbf{E}\left[ r_\theta(T) \right] = \mathbf{E}\left[\sum_{t=1}^T \ell(\theta_t,z_t)- \min_{\theta \in \Theta} \sum_{t=1}^T \ell(\theta,z_t) \right] = \mathbf{E}\left[\sum_{t=1}^T L(\theta_t,q_t) \right]- \mathbf{E}\left[\min_{\theta \in \Theta} \sum_{t=1}^T \ell(\theta,z_t) \right].
> $$
> Notice that we *cannot switch the order of expectation and minimization*, and thus
> $$
> \mathbf{E}\left[\min_{\theta \in \Theta} \sum_{t=1}^T \ell(\theta,z_t) \right] { \color{red} \neq} \mathbf{E}\left[\min_{\theta \in \Theta} \sum_{t=1}^T L(\theta,q_t) \right] \Rightarrow \mathbf{E}\left[ r_\theta(T) \right] { \color{red} \neq} \mathbf{E}[R_\theta(T).
> $$
>
> We have attempted to rectify the aforementioned issue in their analysis.  In doing so, we have to eliminate the $\min$ and $\max$ operators, leading to the bound shown in (19) of our paper. But this bound is not useful, as it cannot be employed to control the optimality gap.
>
> ---
>
> Q2: I’m not sure I understand the novelty of the second algorithm compared with Soma, Gatmiry & Jegelka (2022).
>
> A2: At the first glance, our Algorithm 2, which integrates SMD and Exp3-IX, may seem similar to the approach used by Soma et al. [2022]. However, both the design and analysis of Algorithm 2 are fundamentally different from theirs. We outline the novel aspects below.
>
> 1. It's crucial to realize the non-oblivious nature of the online process when selecting the appropriate algorithm and performing the analysis. In particular, the second player must employ an online algorithm designed for non-oblivious MAB. In contrast, the online algorithms and the analysis used by Soma et al. [2022] do not support the non-oblivious property. Therefore, they fail to bound the optimality gap, as discussed in A1.
>
> 2. The non-oblivious nature, coupled with the stochastic setting, means that existing theoretical guarantees for SMD and Exp3-IX cannot be applied directly. To bound the regret of SMD, we need to make use of the “ghost iterate” technique, as detailed in Section C.3 of the supplementary material. Similarly, when bounding the regret of Exp3-IX, we need to construct new martingale sequences to deal with stochastic rewards, as shown in Section C.4.
> ---
>
> Q3: I think the weakness of the first algorithm is that in many cases, it might be impossible to obtain samples from all distributions.
>
> A3: In that case, we could use Algorithm 2, which only draws 1 sample per iteration, or Algorithm 3, which uses less samples via random sampling.
>
> ---
>
> Q4: Mini-batch sizes in Algorithm 4 can be very large
>
> A4: It only happens when the data is highly imbalanced. In such cases, we can combine the idea of random sampling and mini-batches, which will increase the number of iterations and reduce the size of mini-batches.
>
> ---
>
> Q5: I think it would be better to add some experiments.
>
> A5: We have carried out preliminary experiments, the results of which can be found in the global response to all reviewers. These empirical findings align closely with our theories.

---

> > ### Comment · Reviewer_EKF4 · 2023-08-12
> >
> > I thank the authors for their rebuttal. Regarding the algorithm contribution, thank you for clarifying the issue with Soma et al. [2022]. In my opinion, algorithm 2 is the most interesting but I still think its novelty is limited compared with Soma et al. [2022]. The fix seems quite straightforward from existing techniques for analyzing MAB algorithms. With the additional empirical results included, I'm increasing my score to 5.

---

> > > ### Author Response · Authors · 2023-08-12
> > >
> > > Dear Reviewer EKF4,
> > >
> > > Thank you very much for your kind reply! We will revise our paper according to the constructive reviews. In particular, we will emphasize that Soma et al. [2022] is the *first* work that applies MAB to GDRO, and we will clarify our difference, which mainly lies in the theoretical analysis.
> > >
> > > Best
> > >
> > > Authors

---

### Official Review · Reviewer_Gpdz · 2023-07-04

**Soundness:** 3 good
**Presentation:** 2 fair
**Contribution:** 2 fair
**Rating:** 4
**Confidence:** 4

**Summary:**

The paper presents a comprehensive study on group distributionally robust optimization (GDRO) by formulating it as a convex-concave saddle point problem. The authors propose utilizing mirror descent algorithms and online convex optimization techniques to effectively address this problem. Furthermore, the authors investigate the case of unequal sample sizes drawn from each distribution, leading to the development of a weighted GDRO algorithm. This work provides valuable insights into GDRO and offers practical solutions to address various scenarios.

**Strengths:**

The GDRO problem presented in the paper is interesting and well-defined. It has broad applications and holds significant interest. The authors demonstrate a deep understanding of the literature on mirror descent algorithms and their variants, showcasing their expertise in the field.

**Weaknesses:**

Unfortunately, the paper has several weaknesses, which are listed below:

1) Contribution is unclear: Upon careful review of the paper, I have concerns regarding the lack of clarity regarding the contributions. Section 2 appears to primarily present known results, where the formulation of GDRO as a convex-concave saddle point problem and its solution using stochastic mirror descent are well-established techniques. As a result, Theorem 1 seems to be directly derived from the work by Nemirovski et al. (2009). Similarly, Algorithm 2 appears to be a widely-known algorithm in the online convex optimization (OCO) literature, with known convergence guarantees (Theorem 2).
In this context, Section 3, which focuses on the "weighted GDRO and SA approaches," seems to be the section where the potential contributions lie. However, it appears to be a straightforward extension of Algorithm 1 and Theorem 1, which have already been outlined in Juditsky et al. (2011).
To provide a clearer understanding of the novel and non-trivial aspects of Theorem 3 and Algorithm 3, it would be beneficial for the authors to explicitly highlight the specific advancements or modifications they have introduced. Clarifying the unique features, surprising results, or novel insights that differentiate Theorem 3 and Algorithm 3 from the existing literature will help in addressing the concerns raised regarding the originality and significance of the contributions.

2) Numerical experiments are missing: I am perfectly fine if theoretical NeurIPS papers do not contain numerical experiments. This paper, however, presents an algorithm (or actually  algorithms) for solving a practically relevant GDRO problem. In my opinion, it is absolutely necessary that when presenting algorithms to solve a known problem - these algorithms are numerically verified. Also if something is new about Algorithm 3 (which I am not fully sure - see point above) - I would be curious to see if this algorithm behaves better then existing approaches.

3) Technical issues: See questions in the next paragraph.

**Questions:**

There are several technical questions:
1) Assumption 2 would be much more natural if it was stated as an assumption on the loss function and not the risk.
2) When maxima and minima are taken within the paper - I don’t understand why they exist. For example, in (4) why are maxima and minima attained? Similarly in (7).
3) Where is \bar q needed in Theorem 3? Or is it used at all? If not why does Algorithm 3 return \bar q?

**Limitations:**

The novelty and contribution of this paper, in my understanding, is questionable.

---

> ### Author Rebuttal · Authors · 2023-08-05
>
> Many thanks for the constructive reviews! We provide detailed responses below, and hope the reviewer could reassess the significance of our results. We are looking forward to addressing any further question in the reviewer-author discussion period.
>
> ---
> Q1: Theorem 1 seems to be directly derived from … Nemirovski et al. (2009)
>
> A1: We totally agree. As we detailed in Lines 47, 55 and 147, Algorithm 1 is a routine application of SMD to (3), and Theorem 1 directly follows from Nemirovski et al. [2009]. In fact, we are very *surprised* that recent studies overlooked the fact Nemirovski et al. [2009] had explored the same problem in Section 3.2 of their paper. So, we feel it is necessary to present Algorithm 1 and give credit to Nemirovski et al. [2009].
>
> ---
> Q2: Algorithm 2 is a widely-known … with known convergence guarantees (Theorem 2)
>
> A2: At the first glance, Algorithm 2 might appear to be a straightforward combination of SMD and Exp3-IX. However, the design and analysis of Algorithm 2 present several challenges:
> 1. It's crucial to realize the non-oblivious nature of the online process when selecting the appropriate algorithm and performing the analysis. In particular, the second player must employ an online algorithm designed for non-oblivious MAB. It's worth noting that this is not a straightforward task; previous work [Soma et al. 2022] did not recognize this issue, making their analysis flawed. For details, please refer to **A1 to Reviewer EKF4**.
>
> 2. The non-oblivious nature, coupled with the stochastic setting, means that existing theoretical guarantees for SMD and Exp3-IX cannot be applied directly. To bound the regret of SMD, we need to make use of the “ghost iterate” technique, as detailed in Section C.3. Similarly, when bounding the regret of Exp3-IX, we need to construct new martingale sequences to deal with stochastic rewards, as shown in Section C.4.
>
> In summary, the integration of SMD and Exp3-IX in Algorithm 2 goes beyond mere combination, and involves innovative problem-solving to yield the desired rates.
>
> ---
> Q3: Section 3 …be a straightforward extension of Algorithm 1 and Theorem 1, which … outlined in Juditsky et al. (2011). To provide a clearer understanding of … Theorem 3 and Algorithm 3, it would be …
>
> A3: We believe the reviewer is referring to Theorem 4 and Algorithm 4, rather than Theorem 3 and Algorithm 3. Below, we highlight the novelty of them.
> 1. While Juditsky et al. [2011, §4.3.1] introduced the concept of "scale factors" in solving the Stochastic Semidefinite Feasibility problem, their approach and context are distinct from ours. In their research, the number of samples drawn from different distributions is equal, but the losses can differ and the distributions are heterogeneous. Conversely, in Section 3 of our paper, the number of samples from various distributions may vary, but the losses are identical and the distributions are homogeneous. As such, we cannot directly apply the techniques of Juditsky et al. [2011].
>
> 2. To adapt the concept of "scale factors" to our specific problem, we innovatively utilize mini-batches to reduce the variance of the stochastic gradient. As the sizes of the mini-batches differ across distributions, the variances become disparate as well. The larger the number of samples, the smaller the variance is. Through this approach, we can simulate the effect of heterogeneous distributions, allowing us to incorporate "scale factors" into our methodology. This novel adaptation of existing concepts represents a key aspect of our contribution to the imbalanced scenario.
>
> 3. When analyzing the convergence behavior of Algorithm 4, determining the value of $p_i$ proves to be a significant challenge. It required extensive effort to ascertain the setting in (57), a discovery that led to elegant simplifications in (68) and (69). The advancements of Theorem 4 are the *distribution-dependent* convergence rates, which, to the best of our knowledge, represent a novel contribution to the field of GDRO.
> ---
> Q4: Contribution is unclear
>
> A4: We summarize our contributions below.
> 1. We regard Algorithm 2 as one main contribution, in which we tackle the *non-oblivious* challenge in both the algorithm's design and its theoretical analysis.
> 2. We consider Algorithm 4 as another major contribution, where we integrate mini-batches and “scale factors” in a novel manner, leading to *distribution-dependent* convergence rates.
> 3. We treat Algorithm 3 as a more modest contribution. Nevertheless, it holds value, since its convergence rate complements the Baseline, and it offers a clear motivation for the weighted GDRO formulation.
>
> Undoubtedly, Algorithm 1 is credited to Nemirovski et al. [2009]. We just want to bring this fact to the community's attention.
>
> ---
>
> Q5: Numerical experiments are missing
>
> A5: We have carried out preliminary experiments, the results of which can be found in the global response to all reviewers. These empirical findings align closely with our theories.
>
> ---
> Q6: Assumption 2 … more natural if it was stated as an assumption on the loss and not the risk
>
> A6: While it may seem more intuitive to assume the convexity of the loss function, it's important to note that this assumption is more *stringent* than assuming the convexity of the risk function, as the former implies the latter. Specifically, Assumption 2 underscores that our algorithms and theories can also accommodate cases where the loss function is non-convex, so long as the risk function remains convex.
>
> ---
> Q7: When maxima and minima are taken … why they exist
>
> A7: The reason is because we assume $\mathcal{W}$ is convex and bounded in Line 120. Note that we employ the *same* assumptions as Nemirovski et al. [2009], which are standard in stochastic approximation.
>
> ---
> Q8: Where is $\bar{\mathbf{q}}$ needed in Theorem 3? …
>
> A8: We apologize for the confusion. $\bar{\mathbf{q}}$ is utilized solely within our analysis and is not required to be returned by Algorithm 3.

---

> > ### Comment · Reviewer_Gpdz · 2023-08-11
> > **After reading Rebuttal**
> >
> > I would like to thank the authors for their thoughtful rebuttal. I especially appreciate the work put into presenting preliminary numerical results. Additionally, the clarification of their contributions, combined with the response provided in A1 to Reviewer EKF4, has been helpful.
> > In light of these factors, I am inclined to revise my assessment of the paper's contribution from "poor" to "fair." Consequently, I will adjust my grade by one point to reflect this change.

---

> > > ### Author Response · Authors · 2023-08-11
> > >
> > > Dear Reviewer Gpdz,
> > >
> > > Thank you so much for your kind reply and for adjusting the score! If there is any issue preventing you from giving a positive score, please let us know. We are very happy to provide further responses.
> > >
> > > Best
> > >
> > > Authors

---

### Official Review · Reviewer_evdu · 2023-07-05

**Soundness:** 4 excellent
**Presentation:** 4 excellent
**Contribution:** 4 excellent
**Rating:** 8
**Confidence:** 4

**Summary:**

In this paper, the authors present a systematic investigation of group distributionally robust optimization (GDRO). First, they consider the standard standing where the algorithm can draw samples from all distributions freely. In this case, they demonstrate that stochastic mirror descent (SMD) attains an optimal $O(m \log m/\epsilon^2)$ sample complexity by drawing $m$ samples per iteration. Then, they make use of non-oblivious MAB to reduce the number of samples from m to 1, and obtain the same complexity.

The authors also consider a practical setting where different distributions may have different sample budgets. To this end, they introduce a weighted formulation of GDRO, and develop two stochastic approaches based on non-uniform sampling and mini-batches. By setting the weights appropriately, they are able to establish distribution-dependent convergence rates.

**Strengths:**

1. DRO/GDRO is an important topic in machine learning, and has wide applications. The authors identify technical flaws in existing works [Haghtalab et al., 2022, Soma et al., 2022], and present simple and principled stochastic approaches. i.e., Algorithms 1 and 2. The sample complexities of their methods are nearly optimal, and the idea of applying non-oblivious MAB to GDRO in Algorithms 2 is smart and novel.

2. The new setting in Section 3 is important in practice, since it is common to encounter imbalanced data sets. The formulation of weighted GDRO is interesting, and provides one possible solution. The theoretical guarantee of Algorithm 4 is very strong in the sense that it demonstrates nearly optimal rates for multiple distributions simultaneously.

3. The paper is well-written and acknowledges previous work properly. The authors present their techniques very clearly, and the results are sound to me.

**Weaknesses:**

There are no empirical studies. While theoretical contributions stand on its own right, I would like to suggest the authors to conduct experiments to verify the main theorems.

**Questions:**

1. The authors have mentioned that the work of Haghtalab et al. [2022] suffers a dependency issue. Could you please elaborate on this point? In particular, which part of their analysis is problematic?

2. Is there any other work on GDRO that also investigates the setting in Section 3?

3. When the risk functions have additional structure, such as strongly convex, it is possible to get better sample complexity?

**Limitations:**

The authors have discussed the limitations of their approaches in their remarks. I do not find any potential negative societal impact of this work.

---

> ### Author Rebuttal · Authors · 2023-08-05
>
> Many thanks for the constructive reviews!
>
> ---
>
> Q1: The authors have mentioned that the work of Haghtalab et al. [2022] suffers a dependency issue. Could you please elaborate on this point? In particular, which part of their analysis is problematic?
>
> A1: In the work by Haghtalab et al. [2022], specifically on page 24, the proof following (15) is problematic. This stems from the fact that they *reuse* samples in their algorithm, which causes their stochastic gradients to be *biased*. As a result, they cannot invoke Fact C.3 to derive the final equation on Page 24. Though Haghtalab et al. label their stochastic gradients as “globally unbiased” on Page 6, this characterization is incorrect. As long as samples are reused, the gradient inherently becomes biased. Even within the context of traditional stochastic optimization, we are not aware of any established theory that allows for the reuse of samples across multiple iterations.
>
> ---
>
> Q2: Is there any other work on GDRO that also investigates the setting in Section 3?
>
> A2: We have not identified any studies within GDRO that consider the imbalanced scenario. In the literature, we only find a related work in agnostic federated learning, where Mohri et al. [2019] examine the imbalanced scenario in the *offline* setting. However, their generalization error bound is dominated by the distribution with the fewest samples [Mohri et al., 2019, Corollary 1], and they did not provide distribution-dependent convergence rates.
>
> ---
>
> Q3: When the risk functions have additional structure, such as strongly convex, it is possible to get better sample complexity?
>
> A3: If we adhere to the convex-concave problem in (3), it seems impossible to get a better sample complexity. This is due to the fact that the convergence rate is contingent on both $\mathbf{w}$ and $\mathbf{q}$. For example, if the risk function is strongly convex in $\mathbf{w}$, we know that $\phi(\mathbf{w}, \mathbf{q})$ is also strongly convex in $\mathbf{w}$; However, $\phi(\mathbf{w}, \mathbf{q})$ remains linear in $\mathbf{q}$, making it hard to improve the convergence rate.
>
> On the other hand, it may be possible to improve the sample complexity if we focus on the original problem in (2). This possibility arises from the fact that if all the risk functions exhibit strong convexity, their pointwise maximum will also be strongly convex. We intend to delve into this intriguing aspect in our future research.
>
> ---
>
> Q4: I would like to suggest the authors to conduct experiments to verify the main theorems.
>
> A4: We have carried out preliminary experiments, the results of which can be found in the global response to all reviewers. These empirical findings align closely with our theories.

---

> > ### Comment · Reviewer_evdu · 2023-08-11
> > **The authors have addressed all my concerns well.**
> >
> > I have read the rebuttal as well as other reviews. The authors have addressed all my concerns well. In particular, the presentation of the experiments has strengthened this paper, and I am inclined to increase my score to 8.

---

> > > ### Author Response · Authors · 2023-08-11
> > >
> > > Dear Reviewer evdu,
> > >
> > > Thank you very much for your kind reply! We will revise our paper according to the constructive reviews.
> > >
> > > Best
> > >
> > > Authors

---

### Official Review · Reviewer_41n7 · 2023-07-26

**Soundness:** 3 good
**Presentation:** 3 good
**Contribution:** 2 fair
**Rating:** 6
**Confidence:** 2

**Summary:**

This paper provides algorithms for group distributionally robust optimization.
For the case in which the sample budget for each distribution is identical,
the proposed algorithm achieves nearly optimal sample complexity.
This paper also considers the sample budgets are different for each distribution and provide an algorithm with non-trivial sample-complexity bounds.

**Strengths:**

- This paper is the first to consider a practical setting in which the sample budget varies by distribution.
- The paper is well structured and easy to follow.

**Weaknesses:**

- Given the results of [Soma et al., 2022], the contribution for the uniform-sample-budget setting appears somewhat limited.
- The obtained bounds in this work includes extra $\log m$ factors, unlike ones by [Soma et al., 2022].

**Questions:**

- Can we remove extra $\log m$ factors?
- It is claimed that the $O(1/\sqrt{n_i})$ rate is optimal, which is not obvious to me.
It is understood that for normal (non-distributionally-robust) learning over a single distribution, the optimal rate is $O(1 / \sqrt{n_i})$.
However, the definition of risk indicators employed in this paper (LHS in Theorem 4) differs from the usual.
Therefore, in order to discuss what the optimal rate is, it is necessary to again prove a lower bound on this risk.
- In the paper, one contribution is explained as reducing the number of samples in each iteration.
Though I agree that total sample complexity matters,
it was not clear to me what practical benefit there is in reducing the number of samples per iteration (while keeping the total number of samples the same).
I would appreciate a more detailed explanation of the benefit.

**Limitations:**

The limitations and potential negative societal impact are adequately addressed.

---

> ### Author Rebuttal · Authors · 2023-08-05
>
> Many thanks for the constructive reviews!
>
> ---
> Q1: The obtained bounds in this work includes extra $\log m$ factors, unlike ones by [Soma et al., 2022].
>
> A1: First, it is important to recognize that the original analysis of Soma et al. [2022] is *flawed*. For a detailed explanation, please refer to **A1 to Reviewer EKF4**. Second, after fixing the error in their analysis, we are only able to derive the bound shown in (19) of our paper. Unfortunately, this bound is not useful, as it cannot be employed to control the optimality gap. To put it more precisely, the bound in (19) cannot be directly compared with our upper bounds, such as that in (18), due to the difference in the left-hand side of the inequality. Consequently, although (19) does not contain the $\log m$ factor, it is not stronger than our results.
>
> ---
> Q2: Given the results of [Soma et al., 2022], the contribution for the uniform-sample-budget setting appears somewhat limited.
>
> A2: Below, we summarize the novelty of Algorithm 2 compared with Soma et al. [2022].
> 1. It's crucial to realize the non-oblivious nature of the online process when selecting the appropriate algorithm and performing the analysis. In particular, the second player must employ an online algorithm designed for non-oblivious MAB. In contrast, the online algorithms and the analysis used by Soma et al. [2022] do not support the non-oblivious property. Therefore, they fail to bound the optimality gap.
>
> 2. Although our Algorithm 2 is composed of SMD and Exp3-IX, existing theoretical guarantees for them cannot be applied directly. That is because the non-oblivious nature, coupled with the stochastic setting, makes the analysis much more challenging. To bound the regret of SMD, we need to make use of the “ghost iterate” technique, as detailed in Section C.3 of the supplementary material. Similarly, when bounding the regret of Exp3-IX, we need to construct new martingale sequences to deal with stochastic rewards, as shown in Section C.4.
> ---
>
> Q3: Can we remove extra $\log m$ factors?
>
> A3: We think it is impossible to remove the $\log m$ factor. It's worth noting that even in the context of non-oblivious MAB, we do have an additional $\log m$ factor, and it is conjectured to be unimprovable [Audibert and Bubeck, 2010, Remark 14]. Another evidence is that in a closely related problem—Collaborative PAC Learning [Blum et al., 2017], the lower bound contains a $\log m$ factor.
>
> Reference:
> Jean-Yves Audibert and Sebastien Bubeck. Regret bounds and minimax policies under partial monitoring. Journal of Machine Learning Research, 11:2785–2836, 2010.
>
> ---
>
> Q4: It is claimed that the $O(1/\sqrt{n_i})$ rate is optimal, which is not obvious to me. … However, the definition of risk indicators employed in this paper (LHS in Theorem 4) differs from the usual.
>
> A4: Notice that when $m=1$, the LHS in Theorem 4 becomes the *standard* excess risk. Thus, if we can establish a rate that is faster than $O(1/\sqrt{n_i})$, then applying it to the case $m=1$ means that the excess risk is reduced at a rate faster than $O(1/\sqrt{n_i})$, which is impossible.
>
> ---
> Q5: It was not clear to me what practical benefit there is in reducing the number of samples per iteration (while keeping the total number of samples the same).
>
> A5: It reduces the running time for a single iteration, enhancing the algorithm's real-time capability. In other words, Algorithm 2 can produce an output for every single sample received, whereas Algorithm 1 requires receiving $m$ samples before it can produce a new output.

---

> > ### Comment · Reviewer_41n7 · 2023-08-16
> >
> > Thank you for your kind response. I'd like to maintain my positive score.

---

> > > ### Author Response · Authors · 2023-08-17
> > >
> > > Dear Reviewer 41n7,
> > >
> > > Thank you very much for your kind reply! We will revise our paper according to the constructive reviews.
> > >
> > > Best
> > >
> > > Authors

---

### Author Rebuttal · Authors · 2023-08-07

We have conducted preliminary experiments to evaluate our algorithms. Below, we present the experimental setting and empirical results. All related figures can be found in the accompanying PDF.

## Datasets
We first use a **synthetic** data from Soma et al. [2022]. We set the group number $m=20$, and for each group $i \in [m]$, we draw a model $\mathbf{w}_i^* \in \mathbb{R}^{1000}$ from the uniform distribution over the unit sphere. For distribution $\mathcal{P}_i$, the sample $(\mathbf{x},y)$ is generated with $\mathbf{x} \sim \mathcal{N}(0, I)$ and $y = \mathrm{sign}(\mathbf{x}^\top \mathbf{w}_i^*)$ with probability $0.9$ and $y = -\mathrm{sign}(\mathbf{x}^\top \mathbf{w}_i^*)$ with probability $0.1$.

We also use the **Adult** dataset, which includes attributes such as age, gender, race, and educational background of $48842$ individuals. Following Soma et al. [2022], we construct $m=6$ groups based on the race and gender attributes, where each group represents a combination of {black, white, others} with {female, male}.  We divide this dataset into training and testing subsets, using a 3:1 ratio.

## GDRO with balanced data
For the synthetic data, we will generate the random sample on the fly, according to the protocol above. For the Adult dataset, we will randomly draw samples from each group in the training set. In other words, $\mathcal{P_i}$ is defined as the empirical distribution over the $i$-th group in the training set.

We refer to the method of Sagawa et al. [2020] and our Algorithm 1 by SMD($1$) and SMD($m$) to underscore that they are instances of SMD with $1$ sample and $m$ samples in each iteration, respectively. We denote our Algorithm 2 as Online($1$) to emphasize that it is based on techniques from online learning and uses $1$ sample per iteration. We set $\ell(\cdot;\cdot)$ to be the logistic loss and utilize different methods to train a linear model. In the balanced setting, we use the max risk in (2) as the performance measure. To evaulate the risk value, we will draw a substantial number of samples, and use the empirical average to approximate the expectation.

We first report the max risk with respect to the number of iterations in Fig. 1. We observe that SMD($m$) is faster than Online($1$), which in turn outperforms SMD($1$). This observation is consistent with our theories, since their convergence rates are $O(\sqrt{(\log m)/T})$, $O(\sqrt{m (\log m)/T})$, and $O(m\sqrt{(\log m)/T})$, respectively. Next, we plot the max risk against the number of samples consumed by each algorithm in Fig. 2. As can be seen, the curves of SMD($m$) and Online($1$) are very close, indicating that they share the same sample complexity, i.e., $O(m (\log m)/\epsilon^2) $. By contrast, SMD($1$) has a much higher sample complexity, i.e., $O(m^2 (\log m)/\epsilon^2)$.

## GDRO with imbalanced Data
For the synthetic data, we set the number of samples as $n_i = 800*(21-i)$. Notice that the Adult dataset is inherently imbalanced. So, we can visit each sample in the training set *once* to simulate the imbalanced setting, and the numbers of samples in $6$ groups are $18012, 7920, 1428, 1388, 908, 486$.

Similarly, we label the Baseline mentioned in the first paragraph of Section 3 as SMD($m$). We designate our Algorithms 3 and 4 as SMD$\_\mathrm{r}$ and SMPA$\_\mathrm{m}$ to highlight that the former combines SMD with random sampling and the latter one integrates SMPA and mini-batches. In the imbalanced setting, we will examine how the risk on each individual distribution decreases with respect to the number of iterations. Recall that the total number of iterations of SMD($m$), SMD$\_\mathrm{r}$ and SMPA$\_\mathrm{m}$ are $n_m$, $n_1$ and $n_m/2$, respectively.

Due to the limitation of space, we only present the risk on the first, middle and last distributions (i.e., $\mathcal{P}\_1$, $\mathcal{P}\_{m/2}$ and $\mathcal{P}\_m$) in Fig. 3 and Fig. 4. First, we observe that our SMPA$\_\mathrm{m}$ is faster than both SMD($m$) and SMD$\_\mathrm{r}$ across all distributions, and finally attains the lower risk is most cases. This behavior aligns with our Theorem 4, which reveals that SMPA$\_\mathrm{m}$ achieves a nearly optimal rate of $O((\log m)/\sqrt{n_i})$ for all distributions $\mathcal{P}\_i$, after $n_m/2$ iterations. We also note that on distribution $\mathcal{P}\_1$, although SMD$\_\mathrm{r}$ converges slowly, its final risk is the lowest. This phenomenon is again in accordance with our Theorem 3, which shows that the risk of SMD$\_\mathrm{r}$ on $\mathcal{P}\_1$ reduces at a nearly optimal $O(\sqrt{(\log m)/n_1})$ rate, after $n_1$ iterations. From Fig.3(c) and Fig.4(c), we can see that the final risk of SMD($m$) on the last distribution $\mathcal{P}\_m$ matches that of SMPA$\_\mathrm{m}$. This outcome is anticipated, as they exhibit similar convergence rates of $O(\sqrt{(\log m)/n_m})$ and $O((\log m)/\sqrt{n_m})$, respectively.

---

### Decision · Program_Chairs · 2023-09-21

**Decision:**

Accept (poster)

**Comment:**

This paper makes significant contributions to the field of optimization, particularly in group distributionally robust optimization (GDRO). It introduces novel algorithms with near-optimal sample complexity, matching lower bounds within logarithmic factors. These contributions incorporate online learning techniques, address non-oblivious scenarios, and introduce a novel weighted GDRO formulation, enhancing the versatility and efficiency of GDRO approaches. These results have the potential to extend the utility of GDRO algorithms, making them applicable in a wider range of scenarios.

The reviews highlight several key concerns. One recurring issue is the lack of empirical experiments to validate the proposed algorithms, which could provide essential insights into their practical performance; while the authors have already added preliminary experiments, a more comprehensive experimental section would strengthen the paper. Reviewers also questioned the novelty of certain algorithmic aspects in comparison to prior work, particularly in the context of non-oblivious online learning. Furthermore, concerns were raised about the practicality of obtaining samples from all distributions in certain scenarios and the potential challenges posed by large mini-batch sizes in one of the proposed algorithms.

Overall, despite these concerns, the reviewers generally acknowledge the paper's technical soundness, clear presentation, and significant contributions to robust optimization.  I recommend to accept the paper, given the authors’ commitment to adequately address the remaining concerns and suggestions from the reviewers in their final version.